

# Soil andic properties as powerful factors explaining deep soil organic carbon stocks distribution: the case of a coffee agroforestry plantation on Andosols in Costa Rica

Tiphaine Chevallier[1], Kenji Fujisaki[1], Olivier Roupsard[1,2], Florian Guidat[3], Rintaro Kinoshita[3,5], Elias de
Melo Viginio Filho[3], Peter Lehner[4], Alain Albrecht[1]

[1] Eco&Sols, IRD, CIRAD, INRA, Montpellier SupAgro, Univ Montpellier, Montpellier, France
[2] CIRAD, UMR Eco&Sols, LMI IESOL, B.P. 1386 CP 18524, Dakar, Senegal
[3] CATIE, Tropical Agricultural Centre for Research and Higher Education, 7170 Turrialba, Costa Rica
[4] Cafetalera Aquiares S.A., PO Box 362-7150, Turrialba, 7150, Costa Rica
[5] Research Center for Global Agromedicine, Obihiro University of Agriculture and Veterinary Medecine, Obihiro, Hokkaido
080-8555, Japan

*Correspondence to*: Tiphaine Chevallier (tiphaine.chevallier@ird.fr)

**Abstract.** Soil organic carbon (SOC) constitutes the largest terrestrial carbon stock. Both distribution and variation of SOC
stocks are needed to constitute reference baseline for studies on temporal SOC change. Specifically, in volcanic areas, the
spatial variation of soil andic properties usually explains the spatial variation of topsoil SOC contents, but SOC data for deeper
soil layers are needed. We measured the andic properties (e.g. pyrophosphate extractable aluminium and allophane contents)
and SOC stocks down to 200-cm depth in a 1 km$^2$ micro-watershed covered by Arabica coffee agroforestry in Costa Rica. We
used diffuse reflectance mid-infrared (MIR) spectroscopy to allow a large number of soil analysis. The objectives of our study
were (i) to evaluate MIRS as a low-cost and rapid tool to measure andic properties and SOC in Andosols and (ii) to predict the
level of SOC stocks down to 200-cm depth.

## 1 Introduction

Soil organic carbon (SOC) is a major component of soil fertility and productivity. Being a larger pool of C than vegetation and
the atmosphere combined (Lal, 2004), SOC is a potential sink for atmospheric $CO_2$, especially where it was previously depleted
by land use. Therefore, there is much research needed regarding accurate quantification and spatial pattern of SOC stocks.
Many factors, including soil type, climate, topography, and vegetation biomass control the spatial variation of SOC at different
scales (Batjes, 2014; Jobbagy and Jackson, 2000). At the plot-scale, high spatial variation of SOC may also occur (Gessler et
al., 2000) and may be affected by land use changes (Chevallier et al., 2000). These variations can increase uncertainty when
comparing SOC stocks under different management practices (Costa Junior et al., 2013). Accurate approaches to apprehend



the SOC spatial variation at different scales ranging from farm to plot scales are thus needed to assess SOC stocks, and the

effectiveness of various soil conservation measures to restore SOC.

Andosols represent about 0.84% of the terrestrial soils and store approximately 5% of the global soil C (Matus et al. 2014). These soils derived from volcanic material, present high amount of short range order (SRO) constituents, *i.e.* allophane or imogolite, much higher SOC concentrations than soils containing well crystallized clays (Batjes, 2014; Feller et al., 2001; Torn et al., 1997), high water retention and low bulk densities (Shoji et al. 1996). Andosols can store up to three times as much SOC

as non-Andosol and clear correlations were found between SOC content and allophane content (Basile-Doelsch et al., 2005). Allophane content or allophanic soil could be related with high specific surface areas to stabilize SOC (Chevallier et al. 2010; Beare et al., 2014). Others have not found clear correlations between SOC content and allophane but between SOC content and $Al_p$ (Percival et al., 2000; Shen et al., 2018). These high SOC stocks were usually explained by SOC stabilization against decomposition due to (i) acidic condition, (ii) Al toxicity, (iii) SOC adsorption on the mineral surfaces (Mayer and Xing,

2001), (iv) complexation, precipitation and formation of organo-metal (Al/Fe) complexes, also called Al/Fe humus complexes (Percival et al., 2000; Scheel et al., 2007; Torn et al., 1997), and/or (v) entrapment in the mesoporosity (Mayer, 1994) with a particular network structure (Chevallier et al., 2010; Mayer et al., 2004; McCarthy et al., 2008).

Deep SOC pools have not been considered to contribute much to the exchange of C between soil and the atmosphere and have not been considered in most global C-budgets. However, storage estimation, and dynamics of deep SOC under different

ecosystems received increasing attention (Bounouara et al., 2017; Cardinael et al., 2015; Mathieu et al., 2015; Rasse et al., 2006; Shi et al., 2013). In a global review for tropical regions, SOC stock equals to 616 to 640 PgC in 0-200 cm soil in regards to 384-403 PgC in 0-100 cm (Batjes, 2014). In volcanic regions with a high occurrence of Andosols, the horizontal stratification of SOC stocks was among the lowest and the most variable compared to the other soil types. The ratio of SOC stocks at 0-30 cm to SOC at 0-100 cm for Andosols has been evaluated as 0.48 with a coefficient of variation of 29% (Batjes 2014). Deep

SOC may thus represent large C stocks in Andosols but difficult to predict from the surface SOC stocks. In addition, the vertical distribution of SRO constituents, which have a strong impact on SOC stabilization and stocks, can be complex in areas where (i) active volcano emits intermittent thin ash deposits, (ii) slopes under humid climate govern drainage, *i.e.* Si leaching or accumulation, and thus the formation and persistence of SRO constituents over the crystallized aluminosilicates such as halloysite (Churchman et al. 2016), and the (iii) soil erosion cause movement of soil materials (Zehetner et al., 2003).

Landscape modelling using factors including slope, curvature, or flow accumulation could thus explain the horizontal and vertical distribution of andic properties and SOC stocks in these soil profiles (Gessler et al., 2000; Lin et al., 2016).

Diffuse reflectance spectroscopic analysis in mid-infrared (MIR) regions coupled with chemometrics is under development to assess and progressively to map SOC contents and stocks (Ben-Dor et al., 2009; Clairotte et al., 2016; Nocita et al., 2015; Visacarra Rossel et al., 2016). Mineral soil properties have specific spectral influence, especially in the MIR region, *i.e.*



reflectance at the wavenumbers between 4000 and 400 cm$^{-1}$. In some cases, MIR spectroscopic analysis could replace soil extractions and add to the understanding of the underlying relationships between soil properties and soil chemistry (Janik et al., 1998). However, there are still few quantitative determinations of major soil minerals, comprising clay minerals, and extractable Al, Fe and Si by MIRS (Soriano-Disla et al., 2014). In areas where andic properties spatially varies, measuring andic properties, e.g. ammonium oxalate extractable Al, Si and Fe, and sodium pyrophosphate extractable Al and Fe, and SOC

contents by MIR spectrometry could be convenient and accurate (Kinoshita et al., 2016; McDowell et al., 2012). Furthermore, because absorbance peaks of SRO constituents, as allophane and imogolite specific of Andosols, were near 1000 cm$^{-1}$ and polymerized silicates around 350 cm$^{-1}$ (Parfitt, 2009), we hypothesized that MIR spectroscopy could be appropriate to classify soil material samples as Andosols or non-Andosols, independently of their origin in the soil profile. This classification could be the basis for building different prediction models for SOC contents in the whole soil profile. Bulk density (Bd) is a physical

soil properties determined by soil constituents and their arrangements, *i.e.* the soil structure. Because SRO constituents control soil Bd (Shoji et al. 1996), if SRO constituents can be assessing by MIR spectroscopy, we hypothesised that MIR spectra would contain enough information to predict Bd. The SOC stocks might be then assessed after predicted Bd and SOC contents. Specifically, this study would check that using MIR spectrometry, in an area where andic properties varies, is a cost- and time-effectively tool to classify different soil material according to their andic properties, and to quantify andic properties

(Allophane, extractable Al, Fe and Si content) and SOC stocks for a large number of soil samples from all soil layers.

In a specific volcanic micro-watershed (1 km$^2$) supporting a productive coffee agroforestry system, andic properties predominantly explained the high spatial variation of topsoil SOC contents, ranged from 48 to 172 gC kg$^{-1}$soil at 0-5 cm depth (Kinoshita et al., 2016). In Kinoshita's study, topographic or vegetation covariates did not explain the spatial variation of topsoil SOC content, but andic properties explained. However, there was any information on SOC stocks in deeper soil layers.

The present study was conducted in the same micro-watershed. Our first objective was to verify that MIR spectroscopy could be a useful tool to asses SOC content and stocks in Andosols whatever the soil layer considered. The second objective was to study the variation of SOC in deeper soil layers down to 200 cm. Finally, we apprehended the development of andic properties by the thickness of the andic soil horizons to explain the variability of SOC stocks observed.

## 2 Materials and Methods

### 2.1 Site description

The research site was a 0.9 km$^2$ watershed located in the Central-Caribbean area of Costa Rica. In the Reventazón river basin on the slope of the Turrialba volcano, the research site was located within the Aquiares Coffee Farm (6.6 km$^2$) situated between 83° 44' 39" and 83° 43' 35"W, and between 9° 56' 8" and 9° 56' 35" N. The elevation of the research site ranged from 1020





to 1280 m a.s.l. and the mean slope was of 11.31°. The mean annual precipitation was 3014 mm between 1973 and 2009. The

climate is tropical humid without dry season. The mean annual temperature is 19.5°C. According to Mora-Chinchilla (2000), volcanic avalanche deposits form the geology of the area, which was originally produced by the collapse of a 1.3 km wide strip of the southeastern slope of the andesitic Turrialba volcano. The Turrialba deposits are mainly andesitic volcanic ashes. Soils are the result of a weathering of a more or less homogeneous andesitic parent materials (Meijer and Buurman, 2003). The indications of lava flows, agglomerates, lahars and ashes are also present. Thin ash deposits are still regular, the last one

occurred in mid-2016. The soils are classified as Andosols (Payan et al., 2009) with pH comprising around 5 in surface (Kinoshita et al. 2016) and 6 below 1 m (data not shown).

This research site has been largely studied and described in details (Kinoshita et al. 2016, Gómez-Delgado et al., 2011). Briefly, before the introduction of the coffee in 1975, the watershed was under housing and garden, including a cardamom plantation, all succeeding to a pristine forest. Since 1975, the vegetation since consisted of planted coffee trees (*Coffea arabica L.*, var

Caturra) with *Erythrina poeppigiana* leguminous shade trees at a density of 7.4 trees ha$^{-1}$. The initial planting density of the coffee trees was 6,300 locations ha$^{-1}$. The Aquiares farm is intensively managed with regular pruning and fertilizer application ($214 \pm 44$ kg N ha$^{-1}$ yr$^{-1}$). The farm complies with the Rainforest Alliance$^{TM}$ for its pest and weed management. Weeds are scarce. The yields of green coffee between 1994 and 2011 were about $1375 \pm 341$ kg green coffee ha$^{-1}$ yr$^{-1}$.

## 2.2 Soil sampling

In October to December 2013, 69 soil profiles in the watershed were sampled after removing surface plant residues (Fig. 1). Soil samples were extracted from soil by using a hand-driven steel auger of 5 cm of diameter to a depth of 200 cm or to lithic contact. Each soil profile consisted of four to ten 20 cm thick soil samples, depending on soil depth. The sampling points in the watershed were designed after the study of Kinoshita et al. (2016), taking the spatial variation of SOC content in topsoil into account. Each of the 69 soil profiles were sampled within the inter-row of coffee plants. The collected 598 soil samples

were oven-dried at 40°C for 72 hours and ground using a mortar and pestle to pass through a 200 µm sieve before analysis.

## 2.3 Mid Infra-Red acquisition

Reflectance spectra in the MIR region were acquired at 934 wavenumbers between 4000 and 400 cm$^{-1}$ at 3.86 cm$^{-1}$ interval using a Fourier transform Nicolet 6700 (Thermo Fischer Scientific, Madison, WI, USA). This spectrophotometer was equipped with a silicon carbide source, a Michelson interferometer as dispersive element, and a deuterated triglycine sulfate detector.

Soil samples were placed in a 17-well plate and were scanned using an auto-sampler (soil surface area scanned: ca. 10 mm²). Each MIR spectrum resulted from 32 co-added scans, and the body of the plate (beside wells) was used as reference standard and scanned once per plate (i.e. every 17 samples). Reflectance was converted into absorbance. Twenty wavenumbers were





removed due to often noisy spectrum. MIR spectra were then used in the range from 4000 to 478 cm$^{-1}$ (2500 and 20,909 nm, respectively). The whole spectrum population was composed of 598 mean spectra.

## 2.4 Laboratory soil analysis

Ten soil profiles down to 200-cm depth, i.e. 98 soil samples in total were selected as representative of the total sample set from a spectral viewpoint (Fig. 2). They were also spatially dispersed in the watershed. These 98 soil samples were analyzed for SOC contents and andic properties. The SOC contents were determined with a CHN elemental analyzer (Carlo Erba NA 2000, Milan, Italy). Al, Si, and Fe associated with active amorphous constituents and organo-metal ligands were extracted with ammonium oxalate solution (Blakemore et al., 1981), and noted ($Al_o$, $Si_o$, and $Fe_o$). Soil samples (0.5 g) were shaken for 4 h in the dark with 0.2 M ammonium oxalate and 0.2 M oxalic acid solution (50 mL) at pH 3. After centrifugation and filtration, $Al_o$, $Si_o$, and $Fe_o$ were measured in the filtrate by ICP-AES. Al and Fe associated with soil organic matter was estimated by extraction with sodium pyrophosphate solution (Blakemore et al., 1981) and noted $Al_p$ and $Fe_p$. However, pyrophosphate has been noted to also extract different forms of Fe and to be not specifically relate to organo-Fe complexes (Parfitt and Childs, 1988).

Soil samples (0.5 g) were shaken for 16 h with a 0.1 M sodium pyrophosphate solution (50 mL) at pH 10. After centrifugation and filtration, sodium pyrophosphate extractable $Al_p$ and $Fe_p$ were determined in the filtrate by ICP-AES.

As bulk density (Bd) required tedious field work and may present lower spatial variability than SOC contents, at least in non-Andosols soils (Don et al., 2007), only seven sampling locations were sampled to determine the Bd of each soil layer down to 200 cm depth. These locations were chosen for their spatial (Fig. 1) and spectral (Fig. 2) representation. For logistical reason and stoniness, digging soil pit for bulk density was not possible exactly on the same points as soil sampling, but were at a short distance from the point (< 100 cm). Bulk densities were determined according to the soil core method using a beveled cylinder (98 cm$^3$). Four replicates were sampled per sampling pit and per soil layer. The soil dry mass was determined on soil samples that were oven-dried 48 h at 105 °C. Soil samples were sieved at 2 mm to separate coarse fragments such as stones and living roots. Coarse fragments represented less than 1% of each soil sample mass and were considered negligible. Bd was determined for each sample by dividing the dry mass of soil by the volume of the cylinder. The 4 replicates were averaged to get one Bd by soil sample at a given depth. Finally, 66 Bd (7 soil pits, 10 soil depths unless the soil profile was partial) were obtained.

## 2.5 Spectral data processing

The 98 MIR absorbance spectra were fitted to the measured soil variables ($Al_o$, $Al_p$, $Si_o$, and $Fe_o$ content) using modified partial least square regressions. No mathematical pre-treatments were applied on the spectra. The 4 calibration models that expressed respectively $Al_o$, $Al_p$, $Si_o$, and $Fe_o$ content as a function of sample spectrum, were then applied to predict $Al_o$, $Al_p$, $Si_o$, and $Fe_o$





contents of all samples from their MIR spectrum. The accuracy of the prediction models was determined by cross validation using the leave-one-out method and by computing (i) the coefficient of determination ($R^2$), (ii) the root mean square error (RMSE) between predicted and measured values, and (iii) the ratio of the standard deviation of the value set to RMSE denoted

as RPD. All calculations were done using the Unscrambler 9.7 Software (CAMO Software Inc., Oslo, Norway).

Similarly, the 98 MIR absorbance spectra and SOC contents of the 98 representative soil samples were used to build a calibration model that expressed SOC content (g kg$^{-1}$ soil) as a function of the sample spectrum. Four alternative calibration models were also tested to predict SOC contents of the soil samples. Four alternative models were based on 2 spectral classes, or 2 andic properties classes. The 4 prediction models were built on each of these 4 classes. Two spectral and two andic

properties classes were defined after applying a principal component analysis (PCA) and a K-means clustering on respectively the 598 soil spectra and the 98 measured andic properties ($Al_o$, $Al_p$, $Si_o$, $Fe_o$, $Al_o+0.5\ Fe_o$, Allophane, $(Al_o-Al_p)/Si_o$) (Terra et al., 2018).

The MIR absorbance spectra and Bd of the 66 measured Bd soil samples were used to build a calibration model that expressed Bd as a function of sample spectrum. Bd is a soil physical property related to soil structure and determined at a centimetre

scale. The prediction model built on spectral signature of soil samples passed through a 200 µm-sieve did not pretend to predict Bd *sensu stricto* but the model would use specific vibrational process of the andic constituents as proxies to predict Bd. This model was then applied to the other 532 samples in order to predict their Bd from their MIR spectrum. Because only 66 Bd were measured, we could not build two different models based on clusters as for SOC content.

## 2.6 Data analysis

Andic properties were estimated by:

(i)   the content of organo-Al complexes determined after extraction of Al by pyrophosphate ($Al_p$ in g 100 g$^{-1}$ soil),

(ii)   the content of amorphous Al, Si and Fe determined after oxalate extraction ($Al_o$, $Si_o$, $Fe_o$ in in g 100 g$^{-1}$ soil),

(iii)   the degree of weathering of the volcanic glass determined according to $Al_o+1/2\ Fe_o$ (g 100 g$^{-1}$ soil) (Shoji et al., 1996),

(iv)   the content of Allophane determined by the formula: $Allophane = 100\ Si_o\ /[23.4 - 5.1(Al_o - Al_p) \div Si_o]$ with $Si_o$, $Al_o$, $Al_p$ in g 100 g$^{-1}$ soil (Mizota and Van Reewijk, 1989),

(v)   the atomic Al:Si ratio of soil aluminosilicates calculated according to the formula Al:Si = $(Al_o–Al_p)/Si_o$ (Parfitt and Wilson, 1985). A ratio of Al:Si about 2 was characteristic for the imogolite, proto-imogolite and Al-rich allophane. When inferior to 2, Al:Si ratio signed Si-rich allophane and the presence of significant amounts of crystallized

minerals (Levard et al., 2012).

SOC stocks of samples were calculated with predicted SOC contents and predicted Bd as follows:



$$SOCstock\ (kgC\ m^{-2}) = SOCcontent\ (gkg^{-1}) \times Bd\ (gcm^{-3}) \times thickness\ (cm) \div 100$$

Spectral cluster and soil depth impacts on andic properties, SOC contents, Bd, and SOC stocks were analysed with two-way ANOVA, after checking that ANOVA assumptions were not violated. If the soil depth had an effect on the variable to be explained at $p < 0.05$, Tukey test was used to assess statistical groups. T-test was used to assess the effect of spectral cluster on the variable to be explained for each depth. A random forest regression model was used to evaluate and to order the importance of andic properties, spectral cluster, and soil depth, on SOC contents.

Random forest is a machine learning technique based on randomly built decision trees. At each node, a subset of covariates is also randomly chosen. Random forest was preferred to multiple linear regression method because it allows collinearity between covariates, non-linear relationships between variable to be explained and covariates, and allows the use of both categorical and numeric covariates. %IncMSE was used to assess the relative importance of covariates to explain SOC content variability. For a given covariate, %IncMSE is the difference of mean standard error (MSE) between model with permutation of this covariate and model without permutation. The bigger the %IncMSE, the larger the importance of this covariate in SOC content prediction.

Measured data and correlation among lab measured variables (SOC content and andic properties) can be found in supplementary materials (S1, S2, S3).

Statistical analyses were carried out using R software (R Core Team, 2018). Golden Software Surfer V 8.0 was used to map soil andic properties and SOC stocks. The Digital elevation model (DEM) was created using the 5 m horizontal resolution and 10 m vertical resolution digital terrain model obtained from the TERRA-1998 project (scale ≈ 1: 25000; CENIGA, 1998).

# 3 Results and discussion

## 3.1 Soil andic properties

### 3.1.1 Prediction of soil andic properties

The performance of the quantitative prediction models of extractable Al, Si, Fe on the whole set of soil samples were correct, but the $Fe_p$ contents could not be predicted well (Table 1), probably because of the lack of the specificity of pyrophosphate extraction on Fe forms (Parfitt and Childs, 1988). In previous published studies, MIR spectroscopic prediction for element oxalate extractable were either observed or not respectively in Soriano-Disla et al. (2014) and Misnany et al. (2009). Our results confirmed those of Janik et al. (1998) and indicated that MIRS can be used as a rapid analytical technique to simultaneously estimate the extractable elements (Al, Fe and Si) of Andosols with an acceptable accuracy, according to the criterion RPD>2 (Chang et al., 2001). As imogolite and allophane showed specific peaks in MIR regions (Parfitt, 2009), the correct quantitative prediction for extractable Si, Fe and Al content were likely directly related to the soil mineralogy and not





to specific vibrations of these individual elements at specific wavelengths. All of 598 predicted data for each of the estimation of andic properties were positive except 9 $Al_p$ data, especially in soil profile 14, and 31 in depth (mean= -0.19 ± 0.08 g $Al_p$ 100 $g^{-1}$ soil). By convention, $Al_p$ content for these soil samples were considered at 0 g $kg^{-1}$ soil. MIR spectroscopy seemed promising to predict soil andic properties, but further studies are still needed as few determinations, especially on

pyrophosphate extractable elements, have been reported in literature.

### 3.1.2 Two clusters of soils defined according to their MIRS spectra are retained to describe the variability of andic soil properties in the whole set

The whole population of soil samples (598 samples) and the population of conventionally measured samples (98 samples) were each separated in two clusters. The clustering on the whole population was made according to their MIR spectra. The

clustering on the measured soil sample population was made according to the measured andic properties data. We checked that the clustering method resulted in a same classification of the soil samples. Except for 18 %, all the measured soil samples of a same cluster were classified in a same spectral (square samples were classified in orange in Fig. 3). The mis-grouped samples (circle samples classified in blue in Fig. 3) were all on the border line between the two clusters. That overlaying clustering showed that MIR spectra were dependant of the andic properties of the samples.

We also checked that the spectral clustering defined two soil sample classes with different andic properties. The variation of the andic properties in all the data set was essentially explained by the spectral clustering (Table 2), and secondly by the soil depth except for $Al_o$+ $0.5Fe_o$ and Al:Si ratio where there was no significant correlation with depth. Spectral classification in two cluster classes (named ALL and H) was powerful for organising and describing two categories of different soil materials (Table 3). The ALL cluster (ALL for Allophanic) was rich in organo-Al complexes (0.42 ± 0.12 g $Al_p$ 100 $g^{-1}$ soil), amorphous

Al, Si and Fe (4.5 ± 1.1 g $Al_o$ 100 $g^{-1}$ soil, 1.6 ± 0.4 g $Si_o$ 100 $g^{-1}$ soil, 1.5 ± 0.2 g $Fe_o$ 100 $g^{-1}$ soil) with a $Al_p$:$Al_o$ ratio about 1.0 ± 0.4, and also rich in allophane (15.8 ± 4.4 g allophane 100 $g^{-1}$ soil) with a Al:Si ratio about 2.6 ± 0.2. The H cluster (H for Halloysitic) was poor in organo-Al complexes (0.18 ± 0.11 g $Al_p$ 100 $g^{-1}$ soil), amorphous Al, Si and Fe (1.5 ± 0.8 g $Al_o$ 100 $g^{-1}$ soil, 0.7 ± 0.3 g $Si_o$ 100 $g^{-1}$ soil, 0.9 ± 0.3 g $Fe_o$ $kg^{-1}$ soil) with a highly variable $Al_p$:$Al_o$ ratio about 1.7 ± 1.5, and also poor in allophane (5.3 ± 2.9 g allophane 100 $g^{-1}$ soil) with a Al:Si ratio about 1.7 ± 0.6. We observed a slight decrease of $Al_p$,

$Al_p$:$Al_o$ ratio, $Fe_o$ and allophane content with depth but not for Al:Si ratio and $Al_o$+0.5 $Fe_o$ (Table 3).

The chemical extraction does not give any structural information but it facilitates differentiation between crystalline and poorly crystalline material.  In volcanic ash soils, with $Al_o$+0.5$Fe_o$ >> 2 % and high Al:Si ratio >>2 (Tab. 3), as in the rich andic material (ALL cluster), a dominance of SRO constituents, e.g. Al-rich allophanes, proto-imogolite and imogolite  (Levard et al., 2012), with  nano-spherule structure could be observed (Filimonova et al., 2016). On the contrary, when Si activity in

solution is high with a smaller Al:Si ratio, as in the H cluster, halloysite was suggested (Parfitt et al., 1997). Then the two





clusters represented two different soil materials with likely various proportions of allophane and halloysite. In both material, the high $Al_p$:$Al_o$ ratio revealed that the majority of active Al groups pertained to organo-Al complexes especially in surface. These two types of soil materials were encountered at each soil depth with a predominance of andic or allophanic  horizons in surface (n=57) and halloysite horizon in depth (n=24; Table 3). It confirmed what has been commonly observed that allophane

was associated with the clay mineral halloysite (Ross and Kerr, 1934 quoted in Parfitt 09). Allophane could be weathered to halloysite (Parfitt, 2009; Torn et al., 1997; Wada, 1989), or depending on the chemistry and the regular deposition of the ash, the Si concentration in solution, governed by precipitation and drainage, halloysite and allophane could also be derived directly from volcanic glass, but in different proportion through the soil profile (Parfitt, 2009; Churchman et al. 2016).

Even if the soil reflectance and andic properties varied continuously (Fig. 3), we observed that a MIR spectral classification in

2 clusters was powerful to describe the variation of andic soil properties at the scale of 1 km$^2$. The ALL and H clusters presented differences in MIR spectra (Fig. 4) because allophane and halloysite have a varying degree of hydration and crystallinity. Allophane, a "non-crystalline" or "poorly-crystalline' material, may dehydrate to halloysite, a phyllosilicate crystalline clay (Torn et al., 1997). These difference in mineralogy and proportion in amorphous minerals (Table 3) caused changes in reflectance intensity and absorption features and enabled to distinguish soil samples with different weathering levels.

Especially, the MIR spectra differed at the bands near 3620 cm$^{-1}$ and 3700 cm$^{-1}$, which are characteristic of halloysite (Hidalgo et al., 2010; Fig. 4). The impact of mineral weathering or pedogenesis pathway on MIR spectra of volcanic derived soil samples was clear enough to build soil material (andic vs. halloysitic) clusters using PCA and K-means algorithm based on spectral data distance (Terra et al., 2018). Noteworthy, it was not the case in all context and at global scale (Viscara Rossel et al., 2016). Most of the time, the soil IR spectra represent complex compositional mixtures of soil materials from diverse origin which

impede the association of each soil type with only one spectral class on a global scale. Furthermore, global soil taxonomic classification is also determined by criteria that include parameters with no direct influence on spectral character. When applied to soil materials sharing the same weathering and pedogenesis processes, vis-NIR spectra (Terra et al., 2018) or MIR spectra seem to be useful tool for soil survey and classification. In volcanic areas where soils constituents and the mineralogy variation strongly impact soil properties, MIR spectra seem to be especially well adapted. In the micro-watershed studied, two types of

soil material, the ALL and H clusters, independent of soil depth but with different andic properties were differentiated by their MIR spectra.

### 3.1.3 Three types of soil profiles in the watershed

Each of the 598 soil samples was classified into either ALL and H clusters. However, each soil profile was composed by 10 soil samples either classified among ALL or H cluster. Among the 69 sampled soil profiles, 35 soil profiles (ALL soil profiles)

were only composed of soil samples in ALL cluster, *i.e.* soil is composed of andic materials down to 200 cm; 8 soil profiles



(H soil profiles) were only composed of soil samples in H cluster; and 26 soil profiles (ALL-H soil profiles) were composed of both categories of soil samples (Fig. 5).

Among the 26 ALL-H soil profiles, 20 soil profiles were composed of soil samples classified in ALL cluster in surface and in H cluster in depth. 5 soil profiles were composed of soil samples without a clear classification between ALL and H with depth.
The thickness of the andic material of the ALL-H soil profiles was given by the depth of the apparition of the H material in the soil profile. The thickness of this andic material layer varied from 0 cm in the H soil profiles to 200 cm in the ALL soil profiles A (Fig. 6).

In this micro-watershed, allophanes predominated over halloysite because total annual rainfall was above 3000 mm and altitude was about 1000 m (Parfitt et al. 1983).  A majority of soil profiles (35) were indeed rich in SRO constituent material and
remained rich in Al-rich amorphous materials down to 200-cm depth. The other soil profiles had a decreasing amount of SRO or amorphous materials with depth. The extent to which these amorphous materials were present depend on composition of volcanic ejecta, volcanic eruption frequency, climate, duration of pedogenesis, weathering, and soil erosion (Zehetner et al., 2003). As composition of volcanic ejecta, climate and paleoclimate were identical over the studied micro-watershed of 1 km$^2$, they could not explain the different stages of soil development, and ash, SRO constituents and phyllosilicate minerals
weathering observed. Without a pronounced dry season, the high leaching environment conditions (Gomez-Delgado et al., 2011; Benegas et al., 2014; Welsh et al., 2018) has likely resulted in the formation of andic materials ($Al_o + 0.5 Fe_o > 2\%$) with dominant presence of Al groups (Al:Si >> 2) associated to organo-metal (Al/Fe) complexes and to allophane or imogolite (Fig. 5, ALL soil profile and ALL-H soil profile in surface). These ALL and ALL-H soil profiles which differed by the thickness of their andic material layer, represented 80% of the soil profiles. The other soil profile types (20%) were mainly composed of
halloysite minerals in surface to depth.

Halloysite could be the result of an older soil development from allophane weathering on older tephras. Halloysite also developed in presence of water and in silica-rich environment, such as in buried soil layers receiving silica from the soil layers above or in zones with restricted drainage that prevent leaching (Churchman et al. 2016). The presence of halloysite in subsoil might be explained by Si inputs due to leaching from the recent deposits right above. This transformation was more likely to
occur in rhyolitic tephra, rich in Si, than in basaltic tephras (Dahlgren et al., 2004). The Turrialba deposits were mainly andesitic volcanic ashes and have been rather homogeneous chemically speaking for thousands of years (Meijer and Buurman, 2003) but recent investigations showed rhyolitic composition and multi-stage magma mixing the rhyolite and basaltic andesite end members (Devitre et al., 2018).

In the ALL-H soil profiles, we noted intermediate allophane contents (10 g 100 g$^{-1}$ soil), between the allophane contents in
ALL soil profile (> 15 g 100 g$^{-1}$ soil) and in H soil profile (< 5 g 100 g$^{-1}$ soil) (Fig. 5). This coexistence of allophane and halloysite marked a transition from the allophane-dominated soils to the halloysite-dominated soils. This could be explained



by halloysite formation occuring in microsites within an allophane matrix (Aomine & Wada, 1962 quoted in Parfitt 2009) thanks to differential formation of amorphous and crystalline weathering products with fluctuating leaching and dessication periods controlling different soil solution Si activities (Churchman et al. 2016). This transition zone has been observed in

pedons at different elevations along the slopes of volcanoes in Ecuador in intermediate altitudes (Zehetner et al., 2003). However, the variations in the soil profile type observed here were at much shorter distance than in Zehetner's study. As the Turrialba volcano has still a recent and regular activity with quite regular and homogeneous deposits, we assumed that andic material thickness and composition likely resulted from an integration through time of pedogeomorphic processes: (i)  regular new ash deposits, (ii) deeper weathering of parent materials, (iii) hydrological dynamics, i.e. Si leaching or accumulation, and

(iv) soil erosion (Gessler et al., 2000). In addition, in unstable landscape positions, the erosion of topsoil material could have outcropped the older halloysitic subsoils (H soil profile) and the andic properties were removed along with the erosion of topsoil materials (Zehetner et al., 2003). The topography and its influence on hydrology, pedogenesis and topsoil erosion (Fig. 11b) may be superimposed over the general trend and could contribute to explain the variation of the different soil profiles and andic material thickness in the micro-watershed.

## 3.2 Soil organic carbon contents

### 3.2.1 Soil organic C content predictions

The performance of the models for the prediction of SOC contents built on the whole set yielded better results (higher $R^2$, higher RPD and smaller RMSE) than using two soil groups separately (Table 4). The predictions have been considered accurate according to usual criteria (Chang et al., 2001). The performance of the predictions was comparable to the performance

observed on Hawaiian soil samples by McDowell et al (2012). However, the SOC predictions from this model led to 12 negative SOC contents values, as for the model built on the H cluster differentiated by weak andic measured properties (Table 4). Negative predicted SOC contents have been already seen with an Andosol dataset in Mc Dowell et al. (2012), who noted difficulties to predict low SOC contents. It could be explained by a too low reflectance signature in the soil spectra, making it unable to be detected by global PLS models. Consequently, we preferred the SOC contents prediction on the two soil clusters

differentiated by their MIRS spectra in order to get no negative SOC content predictions. The RMSE between predicted and measured values ranged from 2 to 5 gC kg$^{-1}$ soil (Table 4). For H cluster the RMSE was about 2 gC kg$^{-1}$ soil with a mean of SOC content of 12 gC kg$^{-1}$ soil. For ALL cluster, the RMSE was about 5 gC kg$^{-1}$ with a mean of SOC content of 46 gC kg$^{-1}$.

### 3.2.2 Andic soil properties explain the variation of SOC contents

Soil organic carbon contents were explained by soil depth and clustering based on soil MIR spectra. SOC contents decreased

significantly with soil depth and were much larger in ALL cluster than in H cluster at all depths (Table 5). As already and



commonly observed, andic soil materials displayed much higher SOC contents than soil with crystalline clays. The more allophane is present, the more SOC is concentrated and preserved in soils (e.g. Torn et al. 1997; Chevallier et al. 2010). The concentration of SOC in the andic materials of this study were similar than those observed in the pedogenetic A horizons of Andosols from Costa Rica (Meijer and Buurman, 2003), Ecuador (Zehetner et al. 2003), Martinique (Chevallier et al. 2010),

or Hawai (Torn et al. 1997).

Correlations were observed between all andic properties and SOC contents (Fig. 7, 8). The strongest correlation was found between $Fe_o$ and $Al_p$ and SOC, suggesting a high occurrence of the organo-metal complexes in SOC stabilization and a stronger impact of ferrihydrite on SOC contents at all depth (Fig. 7) than expected after the previous results in the same study on the surface soil, where $Al_p$ was the most correlated soil variable to SOC (0-5 cm; Kinoshita et al., 2016). Oxalate extractable Fe

($Fe_o$) is a reliable proxy of the ferrihydrite content. Involved in the stabilization of SOM in Andosol (Matus et al. 2014; Filimonova et al. 2016; Parfitt et al. 1997), ferrihydrite could dominate the adsorptive surfaces of the SRO constituents in surface as in depth (Kleber et al. 2005). In our study, the preservation of SOC related to ferrihydrite, even in a relatively small amount (0.9-1.3 g $Fe_o$ $100g^{-1}$ soil), seemed to be the major factor explaining SOC content variation in surface and in depth (Fig. 7 and Fig. 8).

The organo-metal complexes, represented in part by $Al_p$, was the second soil variable to explain the variation of SOC (Fig. 8). It has been also pointed out by several other studies (Beare et al., 2014; Huygens et al., 2005; Kinoshita et al., 2016; Takahashi and Dahlgren, 2016). The organo-Al stabilization is not fully understood (Takahashi and Dahlgren, 2016) but some explanations were usually given: Al toxicity for microorganisms, or electrostatic sorption between Al and SOC which limit the accessibility of SOC to microorganisms or their enzymes. The organo-Al complexes were observed to have stronger

protective capacities against SOC degradation compared to allophane (Boudot, 1992; Powers and Schlesinger, 2002), especially in the surface soil layers (Fig. 8) where they were in higher contents (Fig. 5). It should be particularly true in the studied soils, where the ratios $Al_p$:$Al_o$ were higher in surface than in depth and were quite high > 0.7, especially in ALL-H cluster in surface (around 3 in 0-80 cm soil layer). That high $Al_p$:$Al_o$ ratio revealed that the majority of active Al groups pertained to organo-Al complexes and stabilized the organic ligands from the vegetation inputs.

To a lesser extent, the ratio $(Al_o-Al_p)/Si_o$ explained also the SOC content (Fig. 7) and displayed a threshold near 2.5 (Fig. 8). Beyond that threshold, all soil samples were in ALL cluster and the SOC contents were generally high (> 30 gC $kg^{-1}$ soil and up to 100 gC $kg^{-1}$ soil) whatever soil depth. Allophane and $Al_o + 0.5 Fe_o$, seemed to be poor predictors of SOC contents (Fig. 7), in part because the relationship between these SRO constituents and SOC was mediated by soil depth. For a given amount of allophane, soil surface was richer in SOC than soil in depth. The intercept of the regression between allophane and SOC

decreased with soil depth down to zero for soil samples in deep soil layers (Fig. 8). The C inputs by vegetation likely explained these higher SOC contents, along with a slighter higher SOC content variation, in surface (Table 5; Matus et al., 2014).



Although, in agroforestry systems, the introduction of trees in a crop or a plantation can modify the amount and the distribution of organic carbon inputs to the soil at short distances and in both dimensions, vertically and horizontally (Beer, 1988; Cardinael et al., 2018; Payan et al., 2009; Peichl et al., 2006), the SOC contents were poorly correlated with above-ground biomass in
Andosols (Noponen et al., 2013) and mainly controlled by andic properties even in the topsoil (Kinoshita et al., 2016). Furthermore, the decreasing in SOC content with depth in andic materials (ALL cluster) was much less noticeable compared to other studies on Andosols. The SOC contents decreased slightly and smoothly between the soil surface and 200 cm depth (Fig. 5), and remained still high (40 gC kg$^{-1}$ soil) at 200 cm depth. The mineral matrix explained the SOC stabilization even better in depth than in surface (Fig. 8c).

In conclusion, the SOC content distribution in the watershed was mainly controlled by pedogenic processes. The dominant pedogenic processes should be the development of A horizons by andosolization, which was an accumulation of organic matter, derived from vegetation, stabilized by active Al and Fe and a rapid weathering of non-crystalline minerals in volcanic ejecta. Volcanic ejecta did not contain much carbon (2 to 3 mgC g$^{-1}$ ash, data not shown). This was possible due to regular ash deposits from the volcano activity through modern ages (Mora et al., 2014), the volcanic ejecta composition of small particle
size with a glassy nature and a high porosity and permeability, and a permanent humid weather (Ugolini and Dahlgren, 2002). Rapid weathering released Si, Al, and Fe faster than crystalline minerals could form. The soil solution became over saturated. Complexation of humic/colloidal organic substances with metal (organo-metal, Al/Fe, complexes) was especially noticeable at the soil surface and preferential precipitations of metastable non-crystalline materials (allophane, imogolite, ferrihydrite) would occur in the subsoil (Ugolini and Dahlgren, 2002, Kramer et al., 2012). The stabilization of SOC in per-humid Andosol
was thus the results of absorption, complexation and precipitation of organic material with SRO constituents and metals together with the entrapment in the mesoporosity of allophane (Buurman et al. 2007; Chevallier et al., 2010). SOC stabilization could also result from regular soil burial, due to the repeated overlaying of volcanic ashes, explaining the thickness of an andic material layer, which could be similar a young pedogenetic Andosol A horizon. Over time, or in specific location where Si have accumulated, there could be halloysite formation at the expense of their metastable non crystalline constituents (cluster
H). Formation of crystalline minerals with lower surface area and charge density, and with a different nanostructure would cause loss of stabilized OC and lead to smaller SOC contents (Torn et al., 1997). Soil development and mineralogy was a powerful factor to explain SOC content, regardless of the soil depth.

## 3.3 Soil organic carbon stocks

### 3.3.1 Bulk densities predictions

The performance of the model of prediction for Bd built on the whole set was correct with a RMSE around 0.09 for a Bd average on the watershed of 0.8 (Table 6). Such performance was close the Bd predictions of luvisols by near infrared



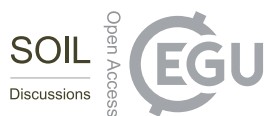

spectroscopy (Cambou et al., 2016). Even if Bd is not only determined by soil constituents but also by soil structure, which has no impact on the MIR spectral signature of a dried and 200 µm-sieved soil sample, the MIR spectra should contain enough information to predict Bd. Andic properties (e.g. $Al_o + 0.5 Fe_o$), well predicted by MIRS, strongly controlled soil Bd (Shoji et

al. 1996) and were used as proxy for Bd prediction by MIRS. Although, the current model of Bd prediction was built on only 66 samples, the MIR spectroscopy applied on soil samples prepared for soil analysis seemed promising to predict Bd of Andosols.

Bulk densities were significantly lower in ALL cluster than in H cluster regardless of depth. Bd increased slightly with soil depth (Table 7) in ALL and H clusters, respectively from 0.65 to 0.75 and to 0.87 to 0.98. As already and commonly observed,

andic soil horizons displayed much lower Bd than soil with crystalline clays (e.g. Mora et al. 2014) independently of soil depth.

### 3.3.2 Variation of soil organic carbon stocks in Aquiares watershed

As for SOC contents, andic soil materials (ALL cluster) showed much higher SOC stocks than soil with crystalline clay materials (H cluster) at every soil depth (Table 7, Fig. 9), as commonly observed (e.g. Mora et al. 2014). We also noticed a smooth decrease of SOC stocks with soil depth (Fig. 9). This decrease was especially smooth and weak in ALL cluster. In this

incrementally accumulated volcanic ash soil, this smooth and weak SOC decrease with depth was possibly due to rejuvenation of the topsoil A horizon by regular fresh ash deposits (Mora et al., 2014) and organic inputs from vegetation.

According to the three soil profile types, SOC stocks down to 2 m depth increased from $24.5 \pm 0.5$ kgC m$^{-2}$ (H soil profile), to $49.9 \pm 1.8$ kgC m$^{-2}$ (ALL-H soil profile) and up to $72.4 \pm 2.0$ kgC m$^{-2}$ (ALL soil profile) in the watershed (Fig. 9). Most of the ecosystem C was then distributed in soils, whereas the above-ground biomass of coffee and *Erythrina poeppigiana* represented

$2.8 \pm 0.2$ kgC m$^{-2}$ only (Charbonnier et al., 2017) and the below-ground biomass of coffee 0.9 kgC m$^{-2}$ (Defrenet et al., 2016). The SOC stocks in ALL soil profiles were much higher than previously reported for Andosol (35.3 kgC m$^{-2}$ with a high coefficient of variation of 61% in a world basis in Batjes 2014). However, larger SOC stocks were also measured in homogeneous deep volcanic-ash soils of Andean ecosystems Ecuador under upper montane forest and under high altitude paramo ($87 \pm 12$ kgC m$^{-2}$; Tonneijck et al., 2010) or in young Hydric Andosols derived from recent volcanic ash (Poulenard

et al., 2003). Our results confirmed that similar stocks could also be measured under plantations at moderate altitudes (1000 m a.s.l.), possibly as a result of organic material accumulation from previous land cover (pristine forest, then households) along with andosolization. Limited by the auger length, we have not measured SOC content in deeper layers than 2 m, which could be high as Zehetner et al. (2003) observed in buried A horizons up to 3 m deep. Future studies were thus needed to evaluate (i) SOC stocks in deeper than 200 cm soil layers, and (ii) the vulnerability of these high SOC stocks to land use change. As

Andosols are known to experience irreversible physical changes on drying, such as losing their specific mesostructure, any





modification of pH or soil moisture dynamics by land use changes could lead to the loss of their exceptional capacity to store high amounts of SOC (Shen et al. 2018).

The relative distribution of SOC stocks with depth, i.e. the ratio of SOC stocks at 0-100 cm to SOC stocks at 0-200 cm, differed according to soil profiles from $0.56 \pm 0.03$ for ALL soil profiles to $0.75 \pm 0.10$ for H soil profiles. As Batjes (2014) already

noticed, this ratio is smaller for Andosols than for other soils with crystalline clays. For H soil profiles, dominated by halloysite, we observed a linear relationship between SOC stock in 0-20 cm and SOC stock in 0-200 cm as already observed in soils without andic properties (Andriamananjara et al., 2016). However, this relationship between SOC stock in 0-20 cm and SOC stock in 0-200 was not occurring for ALL and ALL-H soil profiles (Figure 10a). The thickness of the andic soil layer was a better predictor of SOC stock in 0-200 cm than the SOC stocks in the topsoil (Figure 10b).

As already discussed, the progressive accumulation of organic materials with ashes and the formation of organo-metal complexes and non-crystalline minerals with extensive surface area and variable charge (Torn et al., 1997) likely explained the high SOC stocks. Thus, the thickness of the andic material layer was a good predictor. Nevertheless, when $Al_o$-$Al_p$/$Si_o$ ratios $>> 2$, the variation of the SOC stocks remained high and unexplained ranging between 50 and 74 kgC m$^{-2}$.

In topsoil (0-5 cm), the SOC contents were not explained by the topography nor slopes (Kinoshita et al., 2016), but for the

whole soil profile (0-200 cm) the conditions for the highest SOC storage in the study area were found on gentle slopes (Fig. 11; Mora et al., 2014). Emergence of halloysite soil material could be observed in sharp slope changes with lesser SOC stocks in the soil profile (Fig. 11). Erosion and regular ash deposition modelled landscape, andic A horizon thickness and the distribution of SOC down to 200 cm in the landscape. Pedogeomorphic processes as the lateral redistribution of SOC could also explain the high variation of SOC stocks at fine-scale landscape.

**4 Conclusion**

The 1 km$^2$ watershed showed very high SOC stocks in soils compared to vegetation biomass. The large variation of C stocks in the 0-200 cm soil profile at fine scales (m) were related to the horizontal and vertical variations of the andic properties of these young volcanic soils. Knowing surface SOC stocks provided little information about deep SOC stocks, whereas SOC stocks at 0-200 cm depth were positively correlated with thickness of the young andic A horizon. MIR spectroscopy was

confirmed a convenient and accurate tool to classify the soil material type (andic *vs* non andic soil) whatever the soil layer depth, and to predict SOC contents, bulk density, and SOC stocks. The main conclusion of our study is that ignoring topography and soil pedogenesis introduced a serious weakness in current approaches to regional C stocks evaluation, especially in young volcanic areas where mineralogy controls SOC stabilization and stocks with high degree of variation at fine scale.



## 5 Acknowledgements

Our study was part of the CoffeeFlux observatory, developed by CIRAD (Centre de Coopération Internationale en Recherche Agronomique pour le Développement) and CATIE (Centro Agronómico Tropical de Investigación y Enseñanza). This site contributes to FLUXNET (CR-AqC) and belongs to the SOERE F-ORE-T network observatories with support from Ecofor, Allenvi, and the French national research infrastructure ANAEE-F (http://www.anaee-france.fr/fr/). This work was also supported by AIRD (SAFSE) and the ANR, the French Research Agency through 2 projects, Ecosfix (ANR-2010-STRA- 003-
01) and MACACC (ANR-13-AGRO-0005).

The authors warmly thank the field support in Costa Rica, including Álvaro Barquero and his family, Aléxis Pérez and the support from Cafetalera Aquiares: Alfonso and Diego Robelo, Luis Guillermo Navarro, Manuel Jara and Rafael Acuña Vargas.

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

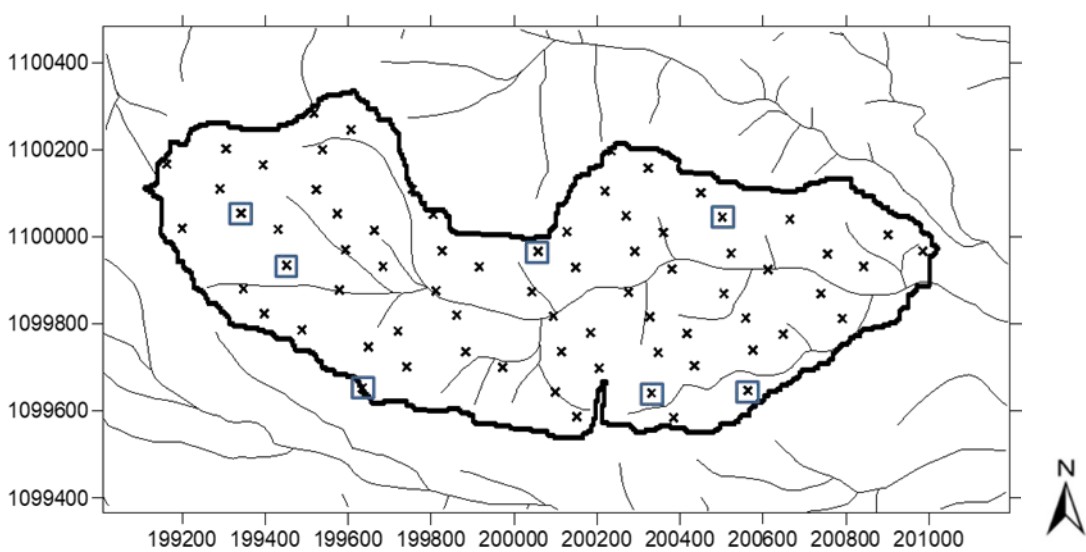

**Figure 1: Map of the sampling 69 locations distributed in the experimental 1 km² micro-watershed with indications of permanent stream channels. See Gómez-Delgado et al. (2011) for more geographical details. The squares indicate the 7 sampling locations for the bulk density measurements down to 2 m depth; x and y are UTM coordinates.**



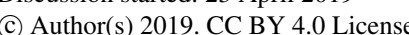

**Figure 2: Principal component analysis of the all soil sample MIR spectra (n=598). Identification of the 98 soil samples (squares) measured conventionally for SOC and oxalate and pyrophosphate extractable Al, Si and Fe contents.**

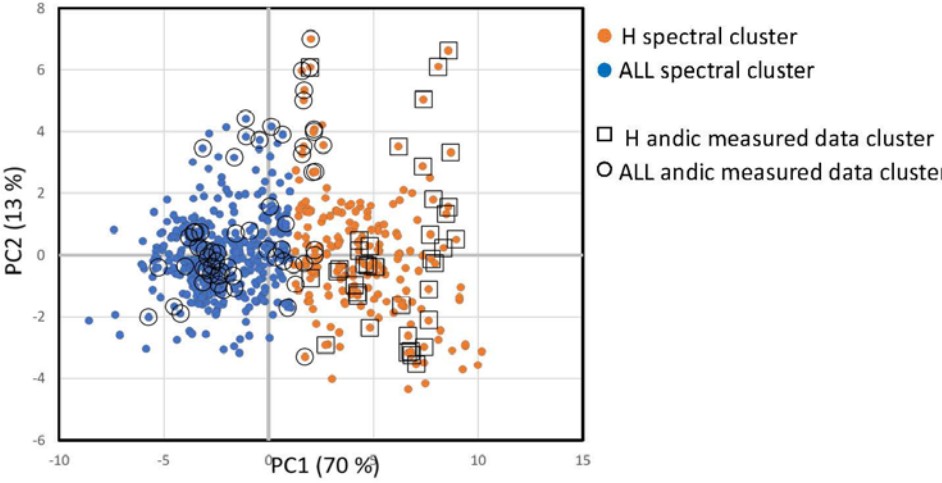

**Figure 3: Principal component analysis (PCA) of the all soil sample MIR spectra (n=598). Identification of two clusters (ALL and H) either built on the PCA of all the 598 MIRS spectra (blue and red points) or on the PCA of the 98 conventionally measured andic soil properties data (circle and squares points).**





**Figure 4: Average mid infra-red spectrum of the soil samples from ALL and H spectral clusters.**



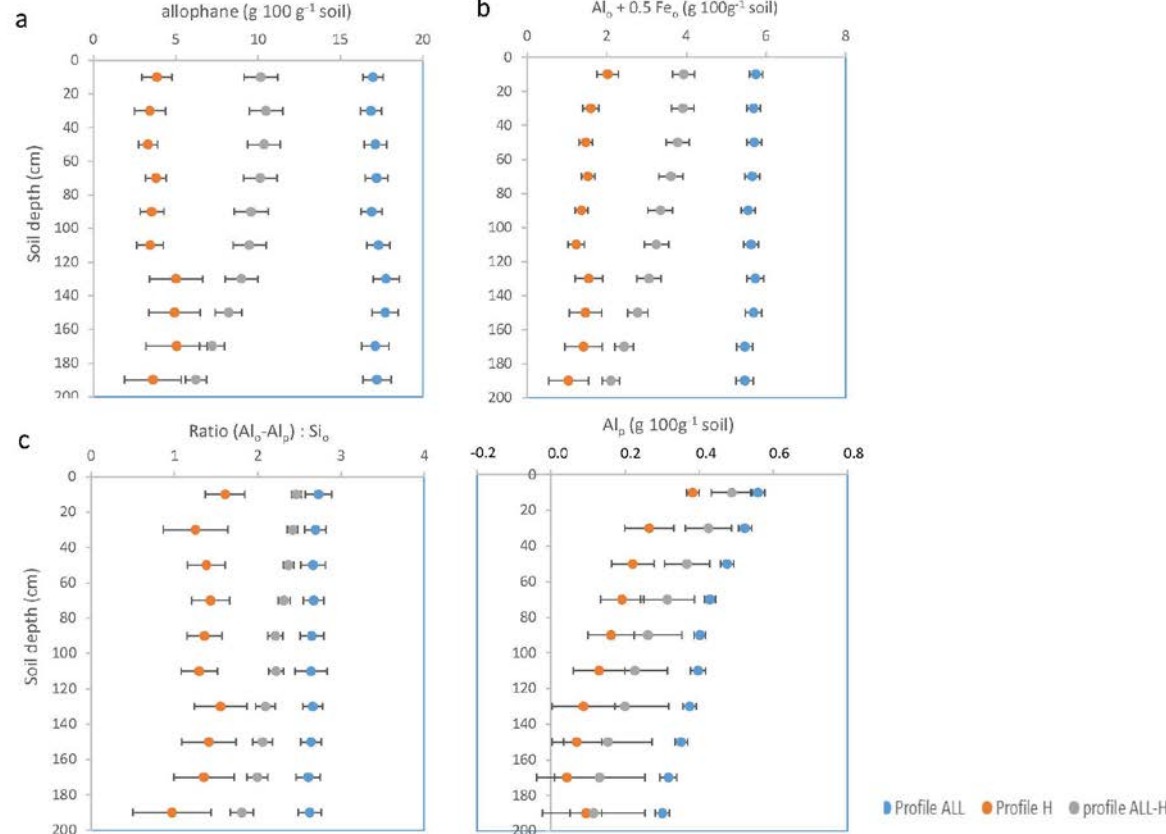

**Figure 5: The three types of soil profile according to four andic properties (a- allophane content, b- Alo + 0.5 Feo indice, c- Al:Si ratio and d- Al-humus or Alp content). ALL soil profile is the mean soil profile of the andic properties of 35 soil profiles; H soil profile is the mean soil profile of the andic properties of 8 soil profiles and ALL-H soil profile is the mean soil profile of the andic properties of 26 soil profiles. Error bars are error types.**



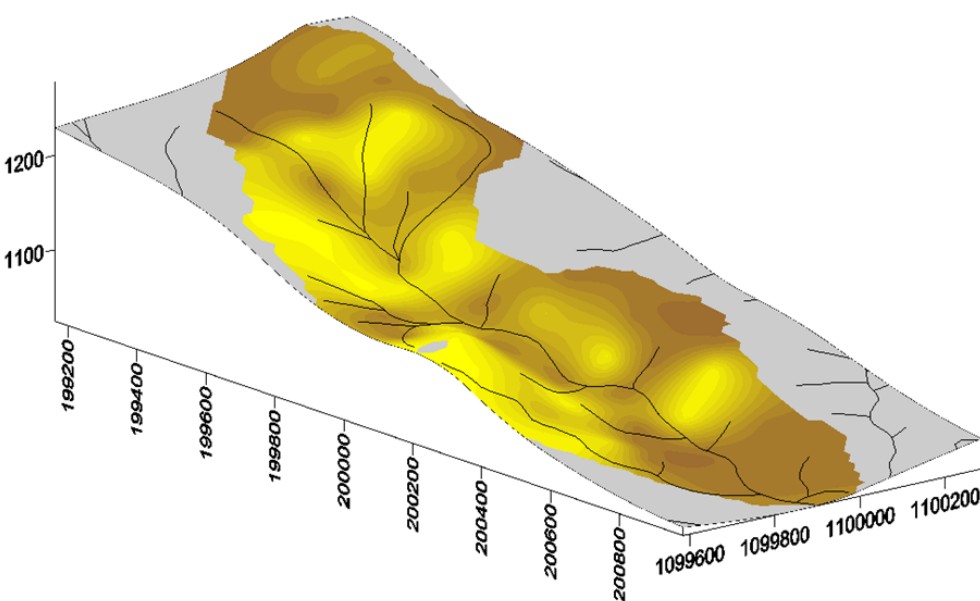

**Figure 6: Map of the thickness of the andic material layer (in cm) in Aquiares watershed. Altitudes above sea level are in m; x and y are UTM coordinates.**

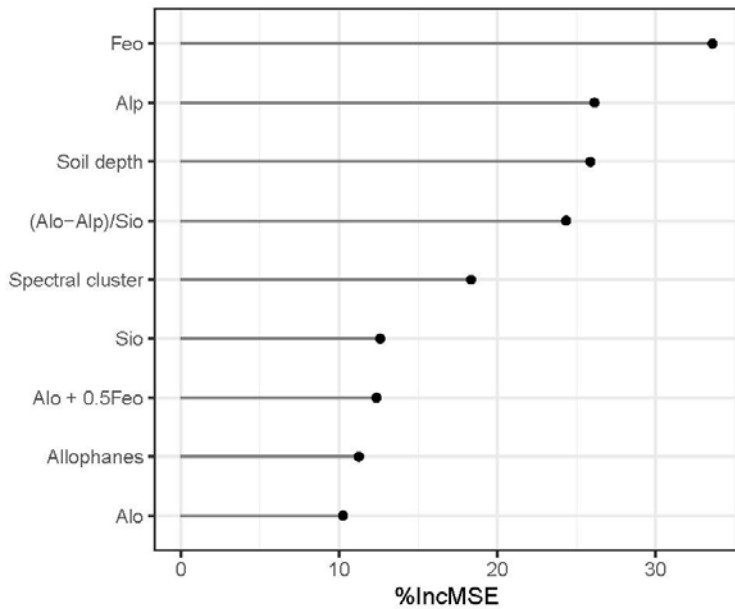

**Figure 7: Relative importance (%IncMSE) of variables in the random forest model for the prediction of SOC content.**





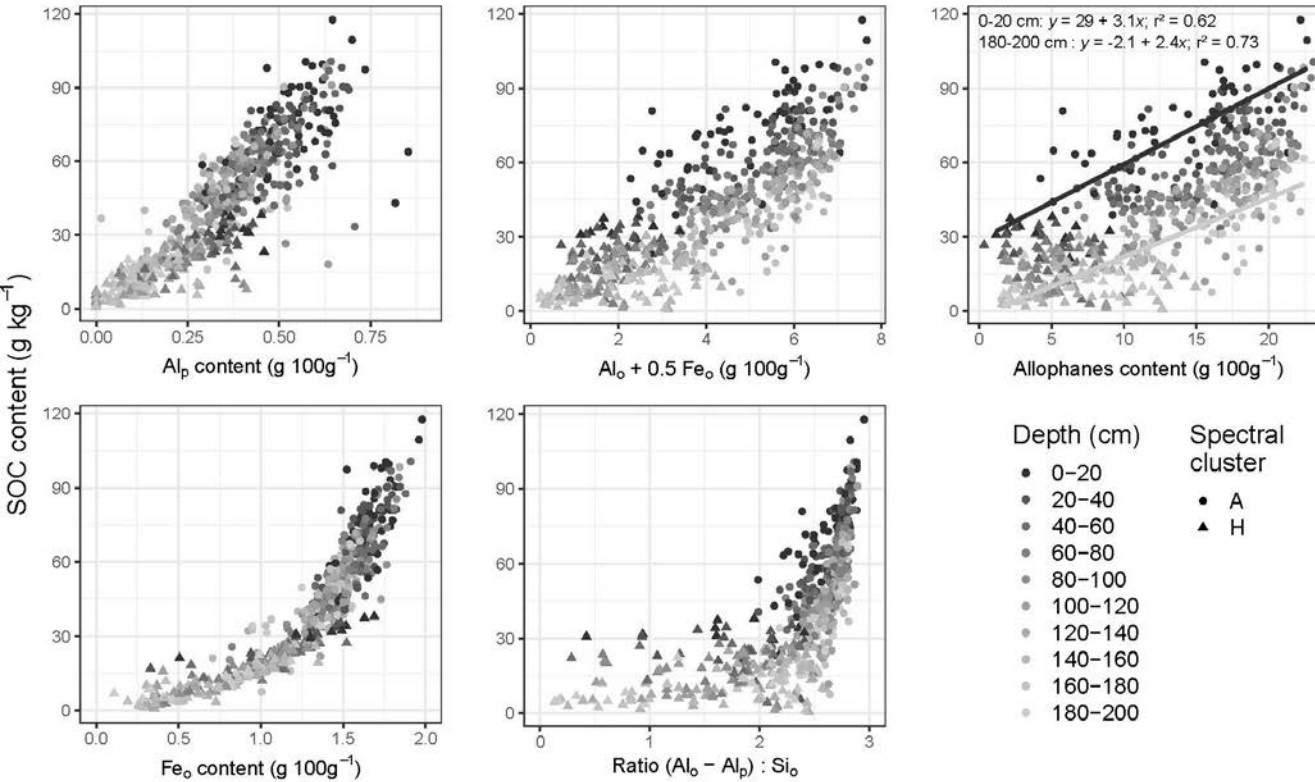

**Figure 8: Relationship between SOC content and andic properties (a- Al-humus or Alp content; b- Alo + 0.5 Feo indice; c- allophane content, 2 linear regressions on surface, 0-20 cm, and on deep, 180-200 cm, soil samples were presented; d- Feo content; e- Al:Si ratio) according to soil sampling depth and spectral cluster classification**



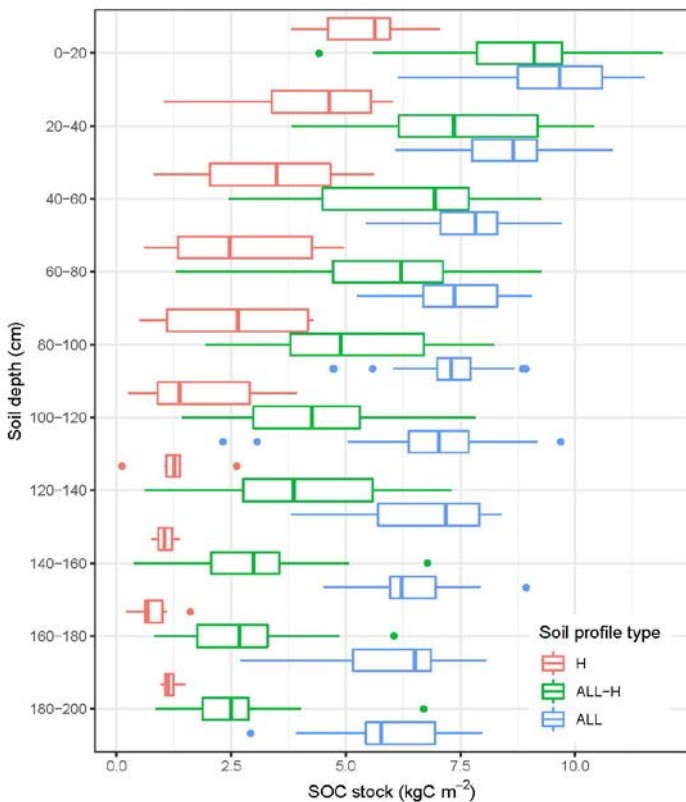

**Figure 9: The SOC stocks soil profiles according to their type of soil profile. ALL Profile is the mean SOC stock soil profile of 35 soil profiles with high andic character; H Profile is the mean is the mean SOC stock soil profile of 8 soil profiles with low andic character and ALL-H Profile is the mean SOC stock soil profile of 26 soil profiles with modification of andic properties along the soil profile.**





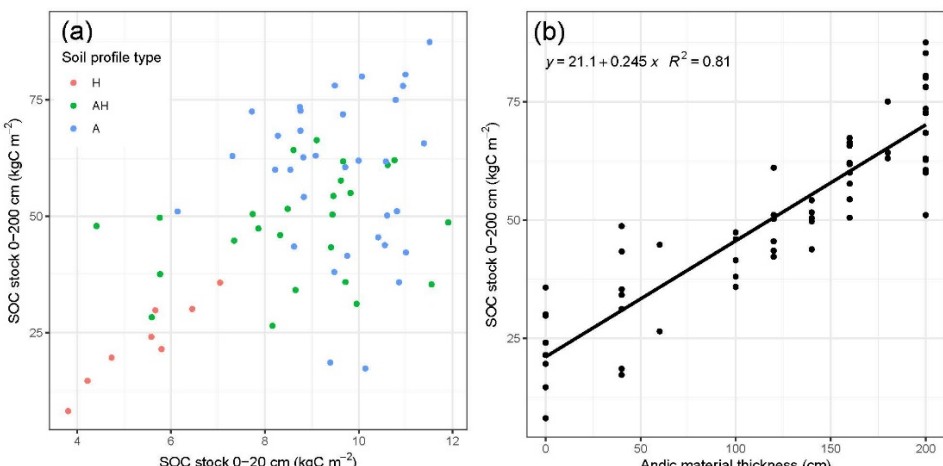

**Figure 10: Relationship between SOC stock at 0-200 cm depth and a) the SOC stock in 0-20 cm depth and b) the thickness of the andic material.**

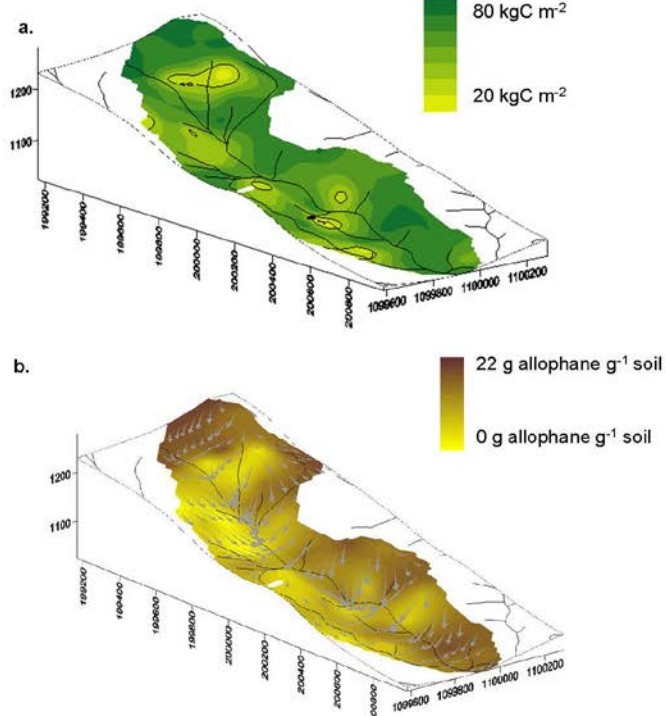

**Figure 11: a- Map of SOC stocks (kgC m⁻²) down to 2 m depth; b- Map of allophane content in the first meter of soil (g 100 g⁻¹ soil) and direction and gradient slopes (arrows). Altitudes above sea level are in m; x and y are UTM coordinates.**





**Table 1: Cross validation statistics of modified partial least square (mPLS) regression for the models of prediction by MIRS of extractable Al, Si and Fe**

|  | n | n outliers | Mean | Sd | RMSE | $R^2$ | RPD |
|---|---|---|---|---|---|---|---|
| $Al_o$ | 98 | 1 | 2.53 | 1.87 | 4.0 | 0.95 | 4.7 |
| $Si_o$ | 98 | 3 | 0.98 | 0.57 | 0.11 | 0.96 | 5.1 |
| $Fe_o$ | 98 | 0 | 1.1 | 0.5 | 0.2 | 0.79 | 2.2 |
| $Al_p$ | 98 | 4 | 0.26 | 0.16 | 0.06 | 0.87 | 2.7 |

n: number of soil samples for calibration. Mean and Standard deviation (Sd) of the measured $Al_o$, $Si_o$, $Fe_o$, $Al_p$ population (in g 100 $g^{-1}$ soil) used for model calibration RMSE: Root Mean Square Error, $R^2$: cross-validation determination coefficient, RPD : ratio of Sd to RMSE, Sd : Standard deviation.


**Table 2: Results of the two way ANOVA, testing the two factors soil depth and spectral cluster class on predicted andic properties and SOC contents**

| Variable | Source | Degree of freedom | Sum of squares | Mean square | F-value | P-Value |
|---|---|---|---|---|---|---|
| $Al_p$ | Cluster | 1 | 7.5 | 7.51 | 832.0 | <0.0001 |
|  | Depth | 9 | 3.4 | 0.38 | 42.0 | <0.0001 |
|  | Residuals | 586 | 5.3 | 0.01 |  |  |
| $Fe_o$ | Cluster | 1 | 50.1 | 50.1 | 1041.0 | <0.0001 |
|  | Depth | 9 | 9 | 1 | 20.9 | <0.0001 |
|  | Residuals | 586 | 28.2 | 0.05 |  |  |
| $(Al_o-Al_p)/Si_o$ | Cluster | 1 | 1698.5 | 1698.51 | 1075 | <0.0001 |
|  | Depth | 9 | 10.6 | 1.18 | 0.747 | 0.666 |
|  | Residuals | 586 | 925.6 | 1.58 |  |  |
| $Al_o + 0.5Fe_o$ | Cluster | 1 | 1478.7 | 1478.68 | 1252.5 | <0.0001 |
|  | Depth | 9 | 9.9 | 1.1 | 0.933 | 0.495 |
|  | Residuals | 586 | 691.9 | 1.18 |  |  |
| Allophanes | Cluster | 1 | 13011.1 | 13011.08 | 1005.1 | <0.0001 |
|  | Depth | 9 | 467 | 51.89 | 4.01 | <0.0001 |
|  | Residuals | 586 | 7586.5 | 12.95 |  |  |






**Table 3. Descriptive statistics of the predicted andic properties of all soil samples (All) with depth and spectral cluster (*ALL* cluster and *H* cluster). Means are followed by standard deviation. Letters indicate significant differences among soil depth at P < 0.05.**

| Depth (cm) | $Al_p$ g 100 $g^{-1}$ soil | | | $Fe_o$ g 100 $g^{-1}$ soil | | | Allophanes g 100 $g^{-1}$ soil | | | $(Al_o\text{-}Al_p)/Si_o$ | | | $Al_o + 0.5 Fe_o$ g 100 $g^{-1}$ soil | | | Number of samples per cluster | | |
|---|---|---|---|---|---|---|---|---|---|---|---|---|---|---|---|---|---|---|
| | All | *ALL* | *H* | All | *ALL* | *H* | All | *ALL* | *H* | All | *ALL* | *H* | All | *ALL* | *H* | All | *ALL* | *H* |
| 0-20 | 0.51±0.12 a | 0.54±0.11 a | 0.37±0.07 a | 1.6±0.21 a | 1.65±0.13 a | 1.33±0.33 a | 12.8±6.2 a | 14.5±5.3 a | 4.3±2.3 ab | 2.5±0.5 | 2.6±0.2 | 1.7±0.6 ab | 4.6±1.7 | 5.1±1.4 | 2.1±0.7 | 68 | 57 | 11 |
| 20-40 | 0.46±0.13 ab | 0.5±0.11 ab | 0.31±0.11 ab | 1.5±0.26 ab | 1.6±0.12 a | 1.15±0.34 ab | 12.9±6.2 ac | 15.3±4.5 ab | 4.1±2.1 b | 2.4±0.6 | 2.6±0.2 | 1.6±0.9 ab | 4.5±1.8 | 5.3±1.2 | 1.9±0.6 | 69 | 54 | 15 |
| 40-60 | 0.41±0.14 bc | 0.46±0.1 bc | 0.24±0.1 bc | 1.43±0.29 bc | 1.56 0.13 ab | 1.05±0.31 abc | 12.9±6.3 ab | 15.8±4.5 ab | 4.6±2.0 ab | 2.4±0.5 | 2.6±0.2 | 1.7±0.6 | 4.5±1.9 | 5.3±1.2 | 1.9±0.7 | 66 | 49 | 17 |
| 60-80 | 0.36±0.12 cd | 0.41±0.08 cd | 0.21±0.1 ce | 1.36±0.3 cd | 1.46±0.17 bc | 0.97±0.28 abcd | 12.8±6.3 ac | 15.7±4.5 ab | 4.4±1.6 ab | 2.4±0.5 | 2.6±0.2 | 1.7±0.5 b | 4.4±1.9 | 5.2±1.2 | 1.8±0.6 | 67 | 50 | 17 |
| 80-100 | 0.32±0.13 de | 0.38±0.09 de | 0.19±0.1 def | 1.29±0.34 de | 1.47±0.18 cd | 0.88±0.26 bcd | 12.4±6.3 ab | 15.9±3.8 ab | 4.4±2.4 ab | 2.3±0.5 | 2.6±0.2 | 1.6±0.5 b | 4.2±1.9 | 5.3±1.1 | 1.7±0.7 | 67 | 47 | 20 |
| 100-120 | 0.3±0.15 de | 0.37±0.11 df | 0.16±0.1 Def | 1.21±0.38 ef | 1.41±0.19 cde | 0.8±0.32 bcd | 12.6±6.5 ab | 16.3±4.0 ab | 4.8±2.3 ab | 2.3±0.5 | 2.6±0.2 | 1.7±0.5 ab | 4.2±2.0 | 5.3±1.1 | 1.8±0.7 | 62 | 42 | 20 |
| 120-140 | 0.27±0.13 ef | 0.34±0.09 df | 0.15±0.09 def | 1.17±0.38 ef | 1.41±0.17 cde | 0.79±0.31 cd | 12.4±6.6 bc | 16.7±3.8 ab | 5.4±3.1 ab | 2.3±0.6 | 2.6±0.1 | 1.7±0.6 ab | 4.1±2.0 | 5.4±1.1 | 1.9±1.0 | 55 | 34 | 21 |
| 140-160 | 0.23±0.13 ef | 0.34±0.09 df | 0.14±0.08 df | 1.11±0.39 fg | 1.4±0.19 cde | 0.84±0.34 cd | 11.9±6.4 b | 17.3±3.9 b | 7.2±3.9 a | 2.2±0.6 | 2.6±0.1 | 1.9±0.7 a | 3.9±2.0 | 5.6±1.1 | 2.4±1.3 | 53 | 25 | 28 |
| 160-180 | 0.2±0.13 f | 0.3±0.11 f | 0.12±0.08 df | 1.00±0.39 g | 1.31±0.19 e | 0.73±0.3 d | 10.8±6.3 bc | 16.6±3.7 ab | 6.0±3.2 ab | 2.2±0.7 | 2.6±0.1 | 1.8±0.7 ab | 3.5±1.9 | 5.3±1.0 | 2.0±1.0 | 48 | 22 | 26 |
| 180-200 | 0.19±0.12 f | 0.31±0.08 ef | 0.1±0.06 f | 0.98±0.38 g | 1.33±0.18 de | 0.71±0.26 d | 10.4±6.5 bc | 16.9±3.8 ab | 5.6±3.0 ab | 2.1±0.7 | 2.6±0.2 | 1.7±0.7 ab | 3.4±2.0 | 5.4±1.0 | 1.9±1.0 | 42 | 18 | 24 |



**Table 4 Cross validation statistics of modified partial least square regression for the models of prediction by MIRS of soil organic C contents (gC kg⁻¹ soil) after calibration on all of the soil samples (All) or on two clusters defined after their MIRS spectra or after their andic properties data.**

|  | n | n outliers | Mean | Sd | $R^2$ | RMSE | RPD | n predicted values <0 |
|---|---|---|---|---|---|---|---|---|
| All samples | 98 | 2 | 33.2 | 23.5 | 0.95 | 5.3 | 4.4 | 12 |
| *ALL* cluster Spectra | 61 | 3 | 46.0 | 20.3 | 0.91 | 5.7 | 3.6 | 0 |
| *H* cluster Spectra | 37 | 2 | 12.0 | 7.6 | 0.92 | 2.1 | 3.6 | 0 |
| *ALL* cluster Andic properties | 31 | 1 | 53.9 | 19.7 | 0.88 | 6.9 | 2.8 | 0 |
| *H* cluster Andic properties | 67 | 3 | 23.6 | 18.4 | 0.93 | 4.6 | 4.0 | 19 |

n number of soil samples, Mean and Standard deviation (Sd) of the measured SOC population used for model calibration. $R^2$, coefficient of regression, RMSE, Root Mean Standard error, RPD, ratio of standard error of prediction and Sd, between measured and predicted SOC contents from the cross validation populations.

**Table 5 Descriptive statistics for SOC contents (g kg⁻¹) of all soil samples (All) with depth and with their spectral cluster (*ALL* and**
***H*); p-value express results from t-test between samples from the two spectra cluster classes (Welsh, two sided alternative). For a given cluster, means followed by the same letters do not differ significantly at p=0.05.**

| SOC g kg⁻¹soil | All | | | *ALL* cluster (398 samples) | | | *H* cluster (199 samples) | | | p-value |
|---|---|---|---|---|---|---|---|---|---|---|
|  | Mean | std | n | Mean | Std | n | Mean | std | n |  |
| 0-20 | 68.0 | 22.5 | 68 | 75.1 f | 16.8 | 57 | 31.2 d | 5.2 | 11 | <0.0001 |
| 20-40 | 58.5 | 21.6 | 69 | 67.3 ef | 14.9 | 54 | 26.8 cd | 7.8 | 15 | <0.0001 |
| 40-60 | 49.8 | 21.3 | 66 | 59.9 de | 14.1 | 49 | 20.9 bc | 6.5 | 17 | <0.0001 |
| 60-80 | 45.4 | 21 | 67 | 54.9 cd | 14.7 | 50 | 17.5 ab | 7.3 | 17 | <0.0001 |
| 80-100 | 41.2 | 20.9 | 67 | 51.8 bd | 14.8 | 47 | 16.2 ab | 6.6 | 20 | <0.0001 |
| 100-120 | 35.8 | 20.2 | 62 | 46.0 abc | 15.7 | 42 | 14.3 ab | 7.6 | 20 | <0.0001 |
| 120-140 | 34 | 21.1 | 55 | 46.8 abc | 15.9 | 34 | 13.3 a | 7.5 | 21 | <0.0001 |
| 140-160 | 28.6 | 19.6 | 53 | 45.6 abc | 14.2 | 25 | 13.5 a | 7.5 | 28 | <0.0001 |
| 160-180 | 24.4 | 17.9 | 48 | 39.5 a | 14.9 | 22 | 11.6 a | 6.7 | 26 | <0.0001 |
| 180-200 | 24.1 | 17.4 | 42 | 41.1 ab | 12.5 | 18 | 11.3 a | 5.1 | 24 | <0.0001 |

**Table 6 Cross validation statistics of modified partial least square regression for the models of prediction by MIRS of soil bulk densities (g cm⁻³)**

| n | n outliers | Mean | Sd | $R^2$ | RMSE | RPD |
|---|---|---|---|---|---|---|
| 62 | 4 | 0.84 | 0.18 | 0.74 | 0.091 | 1.97 |





**Table 7 Descriptive statistics for bulk densities (g cm⁻³) of all soil samples (All) with depth and with their spectral cluster (*ALL* and *H*). P-value express results from t-test between samples from the two cluster classes (Welsh, two sided alternative). For a given cluster, means followed by the same letters do not differ significantly at p=0.05.**

| Bd g cm⁻³ | All | | *ALL* Cluster | | *H* Cluster | | p-value |
|---|---|---|---|---|---|---|---|
| | Mean | Sd | Mean | Sd | Mean | Sd | |
| 0-20 | 0.68 a | 0.14 | 0.65 a | 0.11 | 0.87 | 0.11 | <0.0001 |
| 20-40 | 0.70 a | 0.15 | 0.65 ab | 0.12 | 0.87 | 0.12 | <0.0001 |
| 40-60 | 0.72 ab | 0.16 | 0.66 ac | 0.12 | 0.88 | 0.15 | <0.0001 |
| 60-80 | 0.74 ac | 0.15 | 0.69 ad | 0.12 | 0.89 | 0.13 | <0.0001 |
| 80-100 | 0.77 ad | 0.17 | 0.71 ad | 0.13 | 0.93 | 0.14 | <0.0001 |
| 100-120 | 0.80 cd | 0.16 | 0.73 cd | 0.11 | 0.94 | 0.15 | <0.0001 |
| 120-140 | 0.81 bcd | 0.16 | 0.73 bdc | 0.12 | 0.93 | 0.13 | <0.0001 |
| 140-160 | 0.82 ad | 0.16 | 0.72 ad | 0.10 | 0.91 | 0.14 | <0.0001 |
| 160-180 | 0.87 cd | 0.15 | 0.76 d | 0.11 | 0.96 | 0.12 | <0.0001 |
| 180-200 | 0.88 d | 0.15 | 0.75 cd | 0.11 | 0.98 | 0.09 | <0.0001 |

**Table 8. Descriptive statistics for SOC stocks (kgC m⁻²) of all soil samples (All) with depth and with their spectral cluster (*ALL* and *H*) with depth and with their spectral cluster. P-value express results from t-test between samples from the two cluster classes (Welsh, two sided alternative). For a given cluster, means followed by the same letters do not differ significantly at p=0.05.**

| SOC kgC m⁻² | All | | *ALL* Cluster | | *H* Cluster | | p-value |
|---|---|---|---|---|---|---|---|
| | Mean | Sd | Mean | Sd | Mean | Sd | |
| 0-20 | 8.8 a | 2.0 | 9.4 a | 1.3 | 5.4 a | 1.0 | <0.0001 |
| 20-40 | 7.6 b | 2.0 | 8.5 b | 1.1 | 4.6 ab | 1.4 | <0.0001 |
| 40-60 | 6.6 c | 2.1 | 7.6 c | 1.0 | 3.7 bc | 1.3 | <0.0001 |
| 60-80 | 6.2 cd | 2.1 | 7.2 cd | 1.1 | 3.2 cd | 1.5 | <0.0001 |
| 80-100 | 5.8 bc | 2.2 | 7.0 de | 1.1 | 3.0 cd | 1.3 | <0.0001 |
| 100-120 | 5.3 def | 2.4 | 6.5 def | 1.7 | 2.7 cd | 1.4 | <0.0001 |
| 120-140 | 5.0 def | 2.4 | 6.5 def | 1.4 | 2.5 cd | 1.4 | <0.0001 |
| 140-160 | 4.2 ef | 2.3 | 6.3 def | 1.2 | 2.3 d | 1.3 | <0.0001 |
| 160-180 | 3.8 f | 2.3 | 5.8 e | 1.7 | 2.2 d | 1.2 | <0.0001 |
| 180-200 | 3.8 f | 2.2 | 6.0 ef | 1.3 | 2.1 d | 0.9 | <0.0001 |