# Peer review of "Short-range-order minerals as powerful factors explaining deep soil organic carbon stocks distribution: the case of a coffee agroforestry plantation on Andosols in Costa Rica"

_SOIL, 2019_

## Referee Comment (RC1) · Anonymous Referee #1 · 24 May 2019

I can see that the authors of the study "Soil andic properties as powerful factors explaining deep soil organic carbon stocks distribution: the case of a coffee agroforestry plantation on Andosols in Costa Rica" put some efforts into their study and I found parts interesting. Unfortunately, I also had major problems with the study and I am sorry to suggest that the study should not be published in an international journal.

Comments: 1. Parts of the manuscript are below standard. Unfortunately, important parts of the manuscript are below standard. In the abstract, there is no information on the results of the study. This is unacceptable. The introduction should present the

state of knowledge, open questions, hypotheses and objectives. The authors present two hypotheses which are related to MIRS. One might get the idea that MIRS is the core of the paper. However, half of the introduction (the first three paragraphs) does not deal with MIRS and the remaining part of the introduction does not give specific information on known and open issues with MIRS in the context of the study (which are the known absorption bands, how good was the estimation accuracy in different studies?). Since there are two hypotheses, one would expect two objectives. In fact, there are three objectives, but only one objective deals with MIRS. In summary, there is a need for a major improvement.

2. The core of the study is not sufficiently clear to me. Unfortunately, the core of the study is not sufficiently clear to me. A main focus is on MIRS for which 98 spectra were scanned and where there are also wet-chemical results. In total, however, there are 598 soils and I do not see that the difference (598 - 98, the 500 soils) is really required for this study.

3. The chemometric modelling is not exciting. The authors carried out a LOO-cross-validation for a modified PLS regression using a mixture of replicates and pseudoreplicates (soils from the same profile at different depths). This is not really exciting. More exciting would have been an independent validation, where one makes sure that soils from a profile are kept together in the calibration or validation sample.

4. The anova modelling is not convincing. As above, there are problems with pseudoreplication. In anova, independent data are required. Data from different depths are not independent from each other. Repeated measures anova or a mixed effects model is required.

---

## Referee Comment (RC2) · Anonymous Referee #2 · 5 Jun 2019

Comments on SOIL-2019-14: "Soil andic properties as powerful factors explaining deep soil organic carbon stocks distribution: the case of a coffee agroforestry plantation on Andosols in Costa Rica"

This manuscript reports very interesting results on a highly relevant topic, and the results are based on a large number of soil profile / horizon samples; however, it has room for improvements, both in terms of data evaluation/presentation and writing.

Subsoil OC storage is of top priority in global soil research, and volcanic soils, although

covering a rather small fraction of the land surface, play an important role in global soil OC storage. This study nicely relates the pedogenic development of volcanic soils to their capacity to store OC not only in the topsoil or top 1 meter, but down to 2 m depth, and shows that the mineralogy (i.e. halloysitic vs. short range order) plays a key role in OC stabilization and storage down to 2 m depth. This in itself warrants publication.

The study further explores the use of MIRS to differentiate between the above-mentioned mineralogical groups and to predict resulting effects on soil OC contents and stocks. The presented K-means clustering (Figs. 2 and 3) shows that such differentiation seems possible based on MIRS, albeit not with a sharp line of separation but rather including a "transition" population in between the two clusters. The further prediction of OC contents, bulk density and OC stocks from MIRS spectra seemed to yield accurate results, which shows the potential of this approach for rapid estimation of OC stocks in volcanic soils.

In my opinion, the manuscript currently has several shortcomings, which could all be addressed in a thorough revision: (1) The English is generally comprehensible, but would strongly benefit from revision (with the help of a native speaking soil scientist). (2) The abstract currently does not include any results or conclusions. The study has produced many interesting results, and the authors definitely should include the main results and conclusions in the abstract. (3) The term "andic properties" is not used correctly throughout the manuscript (incl. title, abstract, and esp. in lines 165-175). Andic properties are clearly defined in US soil taxonomy and WRB, and these definitions should be used when claiming that a certain horizon has andic properties or not. The authors have analysed several properties that are specifically important for Andosols (e.g. Alp or Sio or (Alo-Alp)/Sio) but are not "andic properties" per se nor are they requirements for the classification of "andic properties". The authors are advised to be very rigorous in their use of clearly defined terms such as "andic properties", as a misuse of such terms may spread and perpetuate in literature. (4) The accuracy of the developed prediction models was determined by LOO-cross validation using the whole

dataset, if I understood it correctly. While LOO-cross validation is a valid approach for a dataset of independent samples, there may be issues when depth samples of the same profiles (in this study up to 10 depth samples per profile) are included in the dataset. The validation would be much more convincing if it was truly out-of-sample, i.e. using different profiles for calibration and validation, respectively. One way around this could be to perform "LOO-cross validation" leaving out not only one horizon but one entire profile at a time. Alternatively, 75% of the studied profiles could be used for calibration and 25% for validation. Still, the samples of this study originate from a very small (0.9 km2) watershed; the authors need to be aware that even if their models yield accurate predictions for this study area, this may be an indication that similar approaches might work elsewhere, but the parameterized models cannot necessarily be applied directly to other volcanic soils.

Additional comments L198 and elsewhere: "correct" is not the appropriate term for describing the model prediction; "accurate" would be more appropriate. L199: $Fe_p$ is not shown in Table 1. L204-205: What about ferrihydrite? (Fe is not contained in allophane and imogolite) L204-205: delete "and not to specific vibrations..." till end of the sentence. L225-226: To my understanding, Alp/Alo ratios of 1 and higher indicate that (almost) all of the amorphous Al is in the form of organic complexes; but still the authors find 16% allophane in these soils. How can this discrepancy be explained? L234: Did the authors observe nano-spherule structures? If yes, how? L278-280: The second half of this sentence should be rephrased for more clarity. L367: maybe better to replace "weathering" with "formation" L371-378: Rephrase for more clarity. Which are results of this study, which are general statements based on references? L374: replace "would occur" with "occurred" L375: replace "thus" with "likely"; "absorption" or "adsorption"? L398: should be Table 8, not 7. L405-410: If OC stocks are compared to other studies, the depth to which OC stocks were analysed in those other studies needs to be listed, and the comparisons need to address potential differences in depth. L439: "thickness of the young andic A horizon" or "thickness of andic properties"?

Tables and Figures Tables and Figures need to be numbered in the order of their appearance in the text. Figs. 2, 3 and 4 could be combined into one 3-panel figure. Fig. 6 does not have a legend, and I think this figure is not really necessary and could be deleted. Fig. 8d: It seems that there are 2 populations with very different slopes, (roughly) corresponding to the 2 mineralogical groups (halloysitic vs. short range order). Could the different slopes be related to different Fe forms (with different capacity to stabilize OM) in these two types of materials? Fig. 10a: Please use the same abbreviations in the legend as in all other figures. Tables 5, 7 and 8: Please explicitly state in the caption if these tables show statistics of the predicted or measured values (I assume the latter). In Tables 7 and 8, please add the number of replicates (n).

---

## Author Comment (AC1) · 11 Jul 2019

Comment 1 Parts of the manuscript are below standard. Unfortunately, important parts of the manuscript are below standard. In the abstract, there is no information on the results of the study. This is unacceptable. The introduction should present the state of knowledge, open questions, hypotheses and objectives. The authors present two hypotheses which are related to MIRS. One might get the idea that MIRS is the core of the paper. However, half of the introduction (the first three paragraphs) does not deal with MIRS and the remaining part of the introduction does not give specific information

on known and open issues with MIRS in the context of the study (which are the known absorption bands, how good was the estimation accuracy in different studies?). Since there are two hypotheses, one would expect two objectives. In fact, there are three objectives, but only one objective deals with MIRS. In summary, there is a need for a major improvement.

Answers 1/We agreed with the inconsistency of the abstract. The results were in the short abstract and were deleted to the main abstract. We wrote another abstract to underline the need of 598 soil samples (69 soil profiles), the use of MIRS and to give the results of the study.

Here is the new Abstract. Soil organic carbon (SOC) constitutes the largest terrestrial C stock, particularly in the Andosols of volcanic areas. Quantitative information on distribution of SOC stocks is needed to construct a baseline for studying temporal changes in SOC. The spatial variation of soil short-range-order constituents such as allophane usually explains the variability of topsoil SOC contents, but SOC data for deeper soil layers are needed. We found that within a 1-km2 Costa Rican basin covered by coffee agroforestry, SOC stocks in the upper 200 cm of soil were highly variable (24 to 72 kgC m-2). Topsoil SOC stocks were not correlated with SOC stocks present in deeper layers. Diffuse-reflectance mid-infrared (MIR) spectroscopy made possible the analysis of a large number of samples (69 soil profiles, i.e. 598 soil samples) for ammonium-oxalate and sodium-pyrophosphate extractable forms of Al, Fe and Si, as well as SOC content and bulk density. The MIR spectra identified two different soil materials, which occurred one on top of the other in some soil profiles. Andic soil properties and the thickness of a young andic A horizon explained the high variability of SOC. This study illustrates that knowledge of topography and pedogenesis are needed to understand and extrapolate the distribution of SOC stocks at landscape scales.

Short summary. Soil organic carbon (SOC) is the largest terrestrial C stock. Andosols of volcanic areas hold particularly large stocks. E.g. from 24 to 72 kgC m-2 in the upper 2 m of soil, as determined via MIR spectrometry, at our Costa Rican study site:

a 1 km2 basin covered by coffee agroforestry. Andic soil properties explained this high variability, which did not correlate with stocks in the upper 20 cm of soil. Topography and pedogenesis are needed to understand the SOC stocks at landscape scales.

2/ The introduction was rewritten to underline the main objective of our research which was not MIRS but the variability of soil material (andic soil material and halloysite soil material) and then of SOC contents and stocks. Our main objective was to underline that surface SOC stocks does not always give a good image of the SOC stocks in deeper soil horizons in volcanic areas. You need to know about soil pedogenesis if you want to estimate accurately soil carbon stocks in this young volcanic environment. MIRS was an useful tool to analyse a lot of soil samples for andic properties. The only new result in our study about MIRS is to show that MIRS spectra could be used to classify andic material from non andic material.

Here is the new introduction. Soil organic carbon (SOC) not only contributes importantly to soil fertility and productivity, but is a larger pool of C than the world's vegetation and atmosphere combined (Lal, 2004). Those facts suggest that SOC is a potential sink for atmospheric $CO_2$, especially in soils whose formerly high levels of SOC have become depleted through land use. Therefore, many benefits may accrue from quantitative research on spatial patterns of SOC stocks at scales ranging from landscapes down to individual experimental plots. Among the many factors that affect those patterns are soil type, climate, topography, and vegetation biomass (Batjes, 2014; Jobbagy and Jackson, 2000). High spatial variations in SOC content can occur even at small scale (m) due to soil topography (Gessler et al., 2000) or to changes in land use (Chevallier et al., 2000). Such variations increase the uncertainty of comparisons among SOC stocks under different land-management practices (Costa Junior et al., 2013). For example, as when evaluating the effectiveness of different soil-conservation measures for restoring SOC in depleted soils. Attempting to decrease experimental uncertainty by testing SOC at smaller spatial intervals can be impractical because of the time and expense of standard SOC analyses. Therefore, development of accurate, lowcost techniques for quantifying SOC contents at the necessary spatial scales may help researchers carry out experiments that would provide more definitive results. As a soil type on which to test those techniques, Andosols have three attractive traits: they store a disproportionate amount of soil carbon; their SOC patterns are spatially complex both vertically and horizontally (Mora et al., 2014); and the soil constituents (short-range-order constituents, e.g. allophane) associated with SOC in Andosols might be used as proxies for quantifying SOC contents via diffuse-reflectance mid-infrared spectroscopy (MIRS) (Kinoshita et al. 2016). We will discuss each of those traits of Andosols in turn, using the standard nomenclature in which the symbols Alo, Feo and Sio represent ammonium-oxalate-extractable forms of Al, Fe, and Si, and Alp and Fep represent the sodium-pyrophosphate-extractable forms of Al and Fe. Although Andosols represent only about 0.84% of the terrestrial soils, they store approximately 5% of the global soil C (Matus et al. 2014). Derived from volcanic material, they have high levels of short range order (SRO) constituents, i.e. Allophane or imogolite; high SOC concentrations (Batjes, 2014; Feller et al., 2001; Torn et al., 1997); high water retention; and low bulk densities (Shoji et al. 1996). They can store up to three times as much SOC as non-Andosols. Clear correlations have been found between SOC content and Allophane content (Basile-Doelsch et al., 2005) or Aluminium humus complexes (Alp) (Percival et al., 2000; Shen et al., 2018). As explanations for the high SOC contents, most authors have posited that SOC in Andosols is stabilized against decomposition by some combination of (i) acidic condition; (ii) Al toxicity; (iii) SOC adsorption on the mineral surfaces (Mayer and Xing, 2001); (iv) complexation, precipitation, and formation of organo-metal (Al/Fe) complexes, also called Al/Fe humus complexes (Percival et al., 2000; Scheel et al., 2007; Torn et al., 1997); and (v) entrapment in the mesoporosity (Mayer, 1994) with a particular network structure (Chevallier et al., 2010; Mayer et al., 2004; McCarthy et al., 2008). SOC in the deeper levels of Andosols has not been studied extensively. Most calculations of global-level C-budgets have not taken deep SOC into account because SOC at those levels is not considered to contribute much to the exchange of C between soil and the atmosphere. However, authors are now paying increased

attention to dynamics and estimated storage capacities of deep SOC that underlies different ecosystems (Bounouara et al., 2017; Cardinael et al., 2015; Mathieu et al., 2015; Rasse et al., 2006; Shi et al., 2013). In a global review for tropical regions, Batjes (2014) estimated SOC stocks in the 0-200 cm depth range at 616 to 640 PgC, versus 384-403 PgC in the 0-100 cm range. Volcanic regions with high percentages of Andosols (compared to the other soil types) showed some of the lowest degrees of vertical stratification of SOC stocks, but with a high degree of uncertainty. For Andosols, the ratio of SOC stocks at 0-30 cm those at 0-100 cm has been evaluated as 0.48, with a coefficient of variation of 29% (Batjes 2014). According to Churchman et al. (2016), the SRO distribution in volcanic regions can be complex both vertically and horizontally in areas where (i) an active volcano produces thin, intermittent ash deposits, (ii) soil erosion causes movement of soil materials (Zehetner et al., 2003), and (iii) the ash weathers in humid climates on slopes in which zones from which Si is leached away (thereby enabling SRO constituents to form and persist) alternate with zones in which Si accumulates (thereby causing those constituents to crystallize into aluminosilicates like halloysite). Factors (ii) and (iii), especially, can combine to produce terrains in which older, SRO-depleted soils become overlain by newer, SRO-rich topsoils. Previous work (Kinoshita et al., 2016) at the study site described in this article (a 1 km2 volcanic micro-watershed) showed that spatial variations in Alp and allophane contents explained the high spatial variation of SOC contents of topsoils. Those same spatial variations in SOC (from 48 to 172 gC kg-1soil at 0-5 cm depth) were not explained by topographic or vegetation covariates. Kinoshita et al. did not sample deeper layers of soil at the site. However, Kinoshita et al.'s correlations between SRO and SOC, together with Mora et al's (2014) data on the potentially complex distributions of SRO components in volcanic soils, suggested that deep-soil SOC stocks may be not related to topsoil SOC stocks. More generally, we hypothesized that SOC stocks would be highly variable along soil profiles. Testing those hypotheses required the analysis of hundreds of soil samples from soil profiles at widely distributed locations within the site. The results provided a large database with which to test an additional hypothesis:

that signatures of SRO components in MIR spectra of soil samples would be useful proxies for SRO constituents, type of soil material (Andic vs Halloysitic soil material), SOC and bulk density (Bd). Several authors have shown that diffuse-reflectance MIR spectroscopy is a time- and cost-effective analysis to quantify SOC contents. Therefore, MIR spectroscopy has become increasingly popular for spatial mapping of SOC (Ben-Dor et al., 2009; Clairotte et al., 2016; Nocita et al., 2015; Visacarra Rossel et al., 2016). Especially in the MIR region, each of a soil's mineral constituents affects spectra in a characteristic way. For example, absorbance peaks of Allophane and imogolite two SRO constituents that are specific to Andosols are near 1000 cm-1. In contrast, the absorbance peaks of polymerized silicates are near 350 cm-1 (Parfitt, 2009). As SRO constituents control soil Bd (Shoji et al. 1996) and SOC content (Torn et al., 1997), SRO signatures in MIRS spectra may be useful proxies for soil Bd and SOC. Therefore, MIR spectra might contain enough information to predict Bd as well as SOC stocks. Therefore, MIR spectroscopic analysis could replace soil extractions for some purposes (Janik et al., 1998). Those purposes include the spatial mapping of SROs and SOCs in volcanic areas where soil age, type, and andic properties vary. If true, then MIRS could be appropriate for classifying soil samples as Andosols or non-Andosols. Based upon those classifications, researchers could build different prediction models for SOC contents for a large number of soil samples. In summary, then, the three hypotheses that we tested in the work reported here were that 1. Spatial distribution of SOC stocks at depths down to 200 cm depth can vary dramatically volcanic areas, even within a small watershed; 2. Surface SOC stocks in volcanic areas are not reliable predictors of stocks that might exist down to depths of 200 cm; and 3. MIRS is an effective and reliable technique for classifying soil materials according to some characteristics of andic soils associated with contents of SRO constituents (e.g., Alo, Alp, Sio, Feo, Alo+0.5 Feo, Allophane, (Alo-Alp)/Sio).

Comment 2 The core of the study is not sufficiently clear to me. Unfortunately, the core of thestudy is not sufficiently clear to me. A main focus is on MIRS for which 98 spectra were scanned and where there are also wet-chemical results. In total, however, there

are598 soils and I do not see that the difference (598 - 98, the 500 soils) is really required for this study.

Answer: The core of the study and the need of 598 soil samples were specified in the abstract and the introduction, see answer and modifications in the Ms in our answer to comment 1.

Comment 3 The chemometric modelling is not exciting. The authors carried out a LOO-cross-validation for a modified PLS regression using a mixture of replicates and pseudoreplicates (soils from the same profile at different depths). This is not really exciting. More exciting would have been an independent validation, where one makes sure that soils from a profile are kept together in the calibration or validation sample.

Answer: The 10 soil samples of the soil profile sampling were not considered as pseudo-replicates, neither the 69 soil samples of a same depth but from a different soil profile. However, we agree they are not independent, they are spatially linked and were especially used for understanding the vertical and horizontal variability of allophane and carbon contents. We have thus proceeded to independent validations of the prediction models for all Alo, Sio, Feo and Alp, and the global model for SOC content. The models of prediction were built on 7 soil profiles (n=69). Validation was performed on 3 other soils profiles (n= 29.) It was not possible to do it for the other models (Bd, SOC by clusters) as we did not have enough conventionally measured soil samples to constitute two groups. Table 1 and material and method were modified. In this study, the new result is not the model building but the classification of 2 soil materials from the MIR spectra whatever the soil depth of the soil sampling. We underline this information in the introduction (third hypotheses) and it was also discussed in §3.1.2. Clustering according to soil type based upon MIR spectra, versus conventional analyses.

Here are the changes in the text, in Material and Method section (§2.5.1): From the data sets for MIRS (Sect. 2.3) and laboratory analyses (Sect. 2.4.1), we developed predictive models for contents of Alo, Alp, Sio, Feo, and SOC. One model was developed for each constituent, for a total of five models. All of the models were based upon 69 samples, from seven soil profiles, that were common to both data sets. The other 29 samples that were common to both data sets were used for validating the models, as described below. The models were developed by fitting the samples' MIR absorbance spectra to the samples' measured contents of each of the five constituents. Except as noted in Sect. 2.3, the spectra were not given mathematical pre-treatments. Fitting was done via modified partial least-squares regressions. The accuracy of each prediction model was determined by external validation using laboratory analyses and MIRS spectra of the above-mentioned 29 samples, which were from three different soil profiles. The accuracy was quantified by computing (i) the coefficient of determination ($R^2$), (ii) the root mean square error (RMSE) between predicted and measured values, and (iii) the ratio (denoted as RPD) of the standard deviation of the value set to RMSE.

Please see below the new Table 1

Comment 4 The anova modelling is not convincing. As above, there are problems with pseudo replication. In anova, independent data are required. Data from different depths arenot independent from each other. Repeated measures anova or a mixed effects model is required.

Answer : We agreed with the reviewer. As repeated anova was not possible because all the soil profiles were not complete (10 soil depths), linear mixed models were used to analyse our data. Changes were made in the Ms, Changes were also made in table 2, 3, 7 and 8.

Changes in Material and methods §2.6.2: Effects of spectral cluster and soil depth on andic properties, SOC contents, Bd, and SOC stocks were analysed with linear mixed models that considered soil profile a random effect. The T-test was used to assess the effect of spectral cluster on the variable to be explained for each depth. A random forest regression model was used to evaluate and order the importance of andic properties, spectral cluster, and soil depth on SOC contents. Random forest is a machine-learning

technique based on randomly built decision trees. At each node, a subset of covariates is also randomly chosen. Random forest was used instead of multiple-linear-regression methods because it allows use of both categorical and numeric covariates, collinearity between covariates, and non-linear relationships between covariates and the variable to be explained. %IncMSE was used to assess the relative importance of covariates in explaining variability of SOC content. For a given covariate, %IncMSE is the difference between mean standard error (MSE) of the model with permutation of this covariate, and model without that permutation. The larger the %IncMSE, the more important this covariate in predicting SOC content. We used R software (R Core Team, 2018) for the statistical analyses.
* * *
[Figure]

**Table 1. Validation statistics of modified partial least square (mPLS) regression for the models used to predict SOC and extractable Al, Si, and Fe contents from MIR spectra.**

| | | n | n outliers | Mean | Sd | RMSE | $R^2$ | RPD |
|---|---|---|---|---|---|---|---|---|
| $Al_o$ | Calibration | 69 | 0 | 2.18 | 1.72 | 0.4 | 0.94 | 4.9 |
| $Al_o$ | External validation | 29 | | 2.85 | 2.15 | 0.8 | 0.85 | 2.6 |
| $Si_o$ | Calibration | 69 | 3 | 0.88 | 0.53 | 0.14 | 0.93 | 3.9 |
| $Si_o$ | External validation | 29 | | 1.06 | 0.64 | 0.19 | 0.91 | 3.4 |
| $Fe_o$ | Calibration | 69 | 0 | 1.0 | 0.5 | 0.2 | 0.81 | 2 |
| $Fe_o$ | External validation | 29 | | 1.1 | 0.5 | 0.5 | 0.19 | 1 |
| $Al_p$ | Calibration | 69 | 1 | 0.26 | 0.17 | 0.07 | 0.85 | 2.4 |
| $Al_p$ | External validation | 29 | | 0.24 | 0.12 | 0.06 | 0.72 | 2 |
| SOC | Calibration | 69 | 2 | 39.7 | 26.1 | 3.6 | 0.97 | 7.3 |
| SOC | External validation | 29 | | 39.7 | 21.3 | 9.4 | 0.86 | 2.3 |

n: number of soil samples for calibration. Mean and Standard deviation (Sd) of the measured $Al_o$, $Si_o$, $Fe_o$, $Al_p$ population (in g 100 $g^{-1}$ soil) and SOC content (g $kg^{-1}$) used for model calibration RMSE: Root Mean Square Error, $R^2$: cross-validation determination coefficient, RPD : ratio of Sd to RMSE, Sd : Standard deviation.

**Fig. 1.** Table 1: Validation statistics of modified partial least square (mPLS) regression for the models used to predict SOC and extractable Al, Si, and Fe contents from MIR spectra.

---

## Author Comment (AC2) · 11 Jul 2019

This study nicely relates the pedogenic development of volcanic soils totheir capacity to store OC not only in the topsoil or top 1 meter, but down to 2 m depth, and shows that the mineralogy (i.e. halloysitic vs. short range order) plays a key role in OC stabilization and storage down to 2 m depth. This in itself warrants publication.

Answer: Thank a lot for your encouragements. We highlighted the interest of our study in the introduction, see comment 1 in answers to the reviewer 1.

[Figure]

Comment 1 The English is generally comprehensible, but would strongly benefit from revision (with the help of a native speaking soil scientist)

Answer: A English native speaker read our paper and improved English grammar and syntax. See all the Ms and figure caption (on demand)

Comment 2 The abstract currently does not include any results or conclusions. The study has produced many interesting results, and the authors definitely should include the main results and conclusions in the abstract

Answer: The abstract was re-written see the comment 1 to reviewer 1.

Comment 3 The term "andic properties" is not used correctly throughout the manuscript (incl. title, abstract, and esp. in lines 165-175).Andic properties are clearly defined in US soil taxonomy and WRB, and these definitions should be used when claiming that a certain horizon has andic properties or not. The authors have analysed several properties that are specifically important for Andosols (e.g. Alp or Sio or (Alo-Alp)/Sio) but are not "andic properties" per se nor are they requirements for the classification of "andic properties". The authors are advised to be very rigorous in their use of clearly defined terms such as "andic properties", as a misuse of such terms may spread and perpetuate in literature.

Answer: The term of "andic properties" was changed as it was misused for all the analysis of Alp, (Alo-Alp)/Sio etc... We replaced this term by andic soil materials, some characteristics of andic soils, extractable Al, Si, Fe contents or short-range-order constituents content according the context. Note that the title was thus modified to "Short-range-order constituents as powerful factors explaining deep soil organic carbon stocks distribution: the case of a coffee agroforestry plantation on Andosols in Costa Rica"

Comment 4 The accuracy of the developed prediction models was determined by LOO-cross validation using the whole dataset, if I understood it correctly. While LOO-cross validation is a valid approach for a data set of independent samples, there may be

issues when depth samples of the same profiles (in this study up to 10 depth samples per profile) are included in the dataset. The validation would be much more convincing if it was truly out-of-sample, i.e. using different profiles for calibration and validation, respectively. One way around this could be to perform "LOO-cross validation" leaving out not only one horizon but one entire profile at a time. Alternatively, 75% of the studied profiles could be used for calibration and 25% for validation. Still, the samples of this study originate from a very small (0.9km2) watershed; the authors need to be aware that even if their models yield accurate predictions for this study area, this may be an indication that similar approaches might work elsewhere, but the parameterized models cannot necessarily be applied directly to other volcanic soils.

Answer: See comment 3 to reviewer 1. We did not assume that our parameterized model could be applied directly to other volcanic soils. MIRS is used to predict SOC contents, bulk density and SOC stocks for a set of volcanic soils. The new result in our study is not a parameterized model. The new result is to show that the MIRS approach could be an appropriate tool to classify andic from non-andic soil material directly from the soil spectra.

We added a sentence in the text (§3.1.1): MIR spectroscopy seems to be a promising tool for predicting contents of extractable Al, Si, and Fe in volcanic soils, providing that the specific prediction models are based upon conventional analyses of a large set of representative samples.

Additional comments L 198 : correct was replaced by accurate L 199 we deleted the reference to table 1 L 204-205 we deleted Fe to the sentence

L. 225 To my understanding, Alp/Alo ratios of 1 and higher indicate that (almost) all of the amorphous Al is in the form of organic complexes; but still the authors find 16% allophane in these soils. How can this discrepancy be explained?

Answer: We agreed that when allophane content is high, Alp:Alo ratio was small. But our results still show some allophane content ranged from 2 % to up to 10 % for Alp:Alo

about 2. (see supplementary)

Changes in the Ms §3.1.2, L. 277: Surprisingly, and especially in the surface soil, ratios of Alp to Alo are greater than 1, and Allophane contents are high (from 5 to 20 g allophane per 100 g soil). Those results indicate a co-existence of allophane and Al-organo complexes in this ALL material; which may be due to some combination of the soil pH (approximately 5; i.e. near the boundary between allophane and Al/Fe-organo complexes) and the regular inputs of organic materials and ashes from the surface (Mizota and Van Reewijk, 1989).

L. 234 Did the authors observe nano-spherule structures? If yes, how?

Answer: We did not observe nano-spherule structures. We made it clearer in the sentence.

Changes in the Ms §3.1.2, L. 274.The literature (Levard et al., 2012) notes that SRO constituents like Al-rich allophanes, proto-imogolite, and imogolite predominate in volcanic-ash soils with Alo+0.5Feo » 2% and high (»2) Al:Si ratios, such as in the andic-material rich soils in the ALL cluster (Tab. 3). Filimonova et al. (2016) found that the SRO constituents in soils of this type form nano-spherule structures.

L. 367. Absorption was replaced by adsorption. L371-378: Rephrase for more clarity. Which are results of this study, which are general statements based on references?

Answer: We rephrase for more clarity which results came from our study: in last sentence of §3.2.2. Our results showed that deep-soil carbon contents as well as surface-soil carbon contents were essentially driven by the type of soil material and contents of SRO constituents. The combination of soil development and mineralogy was a powerful factor for explaining SOC content, regardless of the soil depth.

L405-410: If OC stocks are compared to other studies, the depth to which OC stocks were analysed in those other studies needs to be listed, and the comparisons need to address potential differences in depth.

Answer: We added the depth of the soil studies cited in the text. Changes in §3.3.2, L. 466. The SOC stocks in this watershed's ALL soil profiles were much larger than the global average reported by Batjes (2014) for Andosols (35.3 kgC m-2, in the upper 2 m of soil, with a high coefficient of variation of 61%). However, larger SOC stocks were also measured in deep, homogeneous volcanic-ash soils of Andean ecosystems in Ecuador under upper montane forest and high altitude paramo (87 ± 12 kgC m-2 within the 0-200 cm depth, Tonneijck et al., 2010), as well as in young Hydric Andosols derived from recent volcanic ash (Poulenard et al., 2003).

L439: "thickness of the young andic A horizon" or "thickness of andic properties"?

Answer: we specified in §3.3.2, L. 486. The thickness of the andic-soil material layer was a better predictor of SOC stock in 0-200 cm than the SOC stocks in the topsoil (Figure 8b).

Fig. 2, 3 and 4 were combined; See Figure 1

Fig. 6 has now a legend. We kept the figure as it shows the variability of the thickness of the andic-soil material layer. It is now the Figure 4

Fig. 8d It seems that there are 2 populations with very different slopes, (roughly) corresponding to the 2 mineralogical groups (halloysitic vs. short range or-der). Could the different slopes be related to different Fe forms (with different capacityto stabilize OM) in these two types of materials?

Answer: We agreed that it seems that there are 2 populations with very different slopes in Fig 8d but could not conclude on the different from of Feo. We added a sentence to notice that result in Section 3.2.2, L. 388: In addition, the relation between SOC content and Feo content appeared to have a threshold near 1.3 g Feo 100g-1 soil. Beyond that threshold, all soil samples were in the ALL cluster. More specifically, the SOC contents were generally high (> 30 gC kg-1 soil) and very sensitive to small variations in Feo content (SOC content = 82 Feo − 66, r2= 0.73). In contrast, SOC contents of soils

below that threshold were lower and less sensitive to Feo content (SOC content = 24 Feo − 5, r2= 0.85).

Fig. 10a we have changed the abbreviations as requested. It is now the Figure 8

Tables 5, 7 et 8 : Please explicitly state in the caption if these tables show statistics of the predicted or measured values(I assume the latter). In Tables 7 and 8, please add the number of replicates (n).

Answer: We explicitly wrote that the tables showed predicted values. The correlations between the measured values were also shown in supplementary materials. The numbers of samples were added in the table.

Modifications: Table 5. Descriptive statistics (predictive data) for SOC contents of all soil samples with depth, and with their spectral cluster (ALL and H); p-value expresses results from t-test between samples from the two spectra cluster classes (Welsh, two sided alternative). For a given cluster, means followed by the same letters do not differ significantly at p=0.05.

Table 7. Descriptive statistics (predictive data) for bulk densities with depth and with their spectral cluster. P-value express results from t-test between samples from the two cluster classes (Welsh, two sided alternative). For a given cluster, means followed by the same letters do not differ significantly at p=0.05.

Table 8. Descriptive statistics (predictive data) for SOC stocks with depth and with their spectral cluster. P-value express results from t-test between samples from the two cluster classes (Welsh, two sided alternative). For a given cluster, means followed by the same letters do not differ significantly at p=0.05.

End of section 2.6.2. Measured data and correlations among laboratory-measured variables (SOC content and the characteristics of andic soils presented above) can be found in the supplementary materials for this article (S1, S2, S3).

Tables and figures were numbered in the order of their appearance.

Please also note the supplement to this comment:
https://www.soil-discuss.net/soil-2019-14/soil-2019-14-AC2-supplement.pdf
* * *
[Figure]

**Fig. 1.** Figure 1. Map of the study site: a 1-km2 micro-watershed. "X" symbols mark the 69 sampling locations. The seven squares show where pits were dug down to a depth of 2 m for bulk-density measurements. T

[Figure]

**Figure 4. Map of the thickness of the andic material layer (in cm) in the Aquiares watershed. The vertical scale indicates elevation (in meters) above sea level. Numbers on x and y axes are UTM coordinates.**

**Fig. 2.** Figure 4. Map of the thickness of the andic material layer (in cm) in the Aquiares watershed. The vertical scale indicates elevation (in meters) above sea level. Numbers on x and y axes are UTM coordi

[Figure]

Figure 8: Relationship between SOC stock at 0-200 cm depth and a) the SOC stock in 0-20 cm depth, b) the thickness of the andic material.

**Fig. 3.** Figure 8: Relationship between SOC stock at 0-200 cm depth and a) the SOC stock in 0-20 cm depth, b) the thickness of the andic material.

**Table 5** Descriptive statistics for SOC contents of all soil samples with depth, and with their spectral cluster (*ALL* and *H*); p-value expresses results from t-test between samples from the two spectra cluster classes (Welsh, two sided alternative). For a given cluster, means followed by the same letters do not differ significantly at p=0.05.

| SOC g kg⁻¹soil | All soil samples | | | *ALL* cluster (398 samples) | | | *H* cluster (199 samples) | | | p-value |
|---|---|---|---|---|---|---|---|---|---|---|
| | Mean | std | n | Mean | Std | n | Mean | std | n | |
| 0-20 | 68.0 | 22.5 | 68 | 75.1 f | 16.8 | 57 | 31.2 d | 5.2 | 11 | <0.0001 |
| 20-40 | 58.5 | 21.6 | 69 | 67.3 ef | 14.9 | 54 | 26.8 cd | 7.8 | 15 | <0.0001 |
| 40-60 | 49.8 | 21.3 | 66 | 59.9 de | 14.1 | 49 | 20.9 bc | 6.5 | 17 | <0.0001 |
| 60-80 | 45.4 | 21 | 67 | 54.9 cd | 14.7 | 50 | 17.5 ab | 7.3 | 17 | <0.0001 |
| 80-100 | 41.2 | 20.9 | 67 | 51.8 bd | 14.8 | 47 | 16.2 ab | 6.6 | 20 | <0.0001 |
| 100-120 | 35.8 | 20.2 | 62 | 46.0 abc | 15.7 | 42 | 14.3 ab | 7.6 | 20 | <0.0001 |
| 120-140 | 34 | 21.1 | 55 | 46.8 abc | 15.9 | 34 | 13.3 a | 7.5 | 21 | <0.0001 |
| 140-160 | 28.6 | 19.6 | 53 | 45.6 abc | 14.2 | 25 | 13.5 a | 7.5 | 28 | <0.0001 |
| 160-180 | 24.4 | 17.9 | 48 | 39.5 a | 14.9 | 22 | 11.6 a | 6.7 | 26 | <0.0001 |
| 180-200 | 24.1 | 17.4 | 42 | 41.1 ab | 12.5 | 18 | 11.3 a | 5.1 | 24 | <0.0001 |

**Table 7** Descriptive statistics for bulk densities with depth and with their spectral cluster. P-value express results from t-test between samples from the two cluster classes (Welsh, two sided alternative). For a given cluster, means followed by the same letters do not differ significantly at p=0.05.

| Bd g cm⁻³ | All soil samples | | *ALL* Cluster (398 soil samples) | | *H* Cluster (199 soil samples) | | p-value |
|---|---|---|---|---|---|---|---|
| | Mean | Sd | Mean | Sd | Mean | Sd | |
| 0-20 | 0.68 f | 0.14 | 0.65 a | 0.11 | 0.87 c | 0.11 | <0.0001 |
| 20-40 | 0.70 ef | 0.15 | 0.65 de | 0.12 | 0.87 bc | 0.12 | <0.0001 |
| 40-60 | 0.72 ef | 0.16 | 0.66 de | 0.12 | 0.88 bc | 0.15 | <0.0001 |
| 60-80 | 0.74 de | 0.15 | 0.69 cd | 0.12 | 0.89 bc | 0.13 | <0.0001 |
| 80-100 | 0.77 cd | 0.17 | 0.71 bc | 0.13 | 0.93 abc | 0.14 | <0.0001 |
| 100-120 | 0.80 abc | 0.16 | 0.73 ab | 0.11 | 0.94 abc | 0.15 | <0.0001 |
| 120-140 | 0.81 abc | 0.16 | 0.73 ab | 0.12 | 0.93 ab | 0.13 | <0.0001 |
| 140-160 | 0.82 bc | 0.16 | 0.72 ab | 0.10 | 0.91 abc | 0.14 | <0.0001 |
| 160-180 | 0.87 ab | 0.15 | 0.76 a | 0.11 | 0.96 a | 0.12 | <0.0001 |
| 180-200 | 0.88 a | 0.15 | 0.75 ab | 0.11 | 0.98 a | 0.09 | <0.0001 |

**Table 8.** Descriptive statistics (predictive data) for SOC stocks with depth and with their spectral cluster. P-value express results from t-test between samples from the two cluster classes (Welsh, two sided alternative). For a given cluster, means followed by the same letters do not differ significantly at p=0.05.

| SOC kgC m⁻² | All soil samples | | *ALL* Cluster (398 soil samples) | | *H* Cluster (199 soil samples) | | p-value |
|---|---|---|---|---|---|---|---|
| | Mean | Sd | Mean | Sd | Mean | Sd | |
| 0-20 | 8.8 a | 2.0 | 9.4 a | 1.3 | 5.4 a | 1.0 | <0.0001 |
| 20-40 | 7.6 b | 2.0 | 8.5 b | 1.1 | 4.6 b | 1.4 | <0.0001 |
| 40-60 | 6.6 c | 2.1 | 7.6 c | 1.0 | 3.7 c | 1.3 | <0.0001 |
| 60-80 | 6.2 cd | 2.1 | 7.2 c | 1.1 | 3.2 cd | 1.5 | <0.0001 |
| 80-100 | 5.8 de | 2.2 | 7.0 cd | 1.1 | 3.0 d | 1.3 | <0.0001 |
| 100-120 | 5.3 ef | 2.4 | 6.5 de | 1.7 | 2.7 de | 1.4 | <0.0001 |
| 120-140 | 5.0 efg | 2.4 | 6.5 de | 1.4 | 2.5 ef | 1.4 | <0.0001 |
| 140-160 | 4.2 fg | 2.3 | 6.3 e | 1.2 | 2.3 ef | 1.3 | <0.0001 |
| 160-180 | 3.8 g | 2.3 | 5.8 e | 1.7 | 2.2 f | 1.2 | <0.0001 |
| 180-200 | 3.8 g | 2.2 | 6.0 e | 1.3 | 2.1 f | 0.9 | <0.0001 |

**Fig. 4.** Tables 5, 7, 8

**Supplement:**

---

## Author Comment (AC3) · 11 Jul 2019

Answer to the comment 4 of the reviewer 1: See the modifications in tables 2, 3, 7 and 8

[Figure]

**Table 2. Results of the linear mixed model testing the effect of soil depth and spectral cluster class upon the predicted andic properties.**

| Variable | Source | Degree of freedom | F-value | P-Value |
|---|---|---|---|---|
| $Al_p$ | Intercept | 1 | 1387.1 | <0.0001 |
| | Cluster | 1 | 561.4 | <0.0001 |
| | Depth | 9 | 81.0 | <0.0001 |
| $Fe_o$ | Intercept | 1 | 3404.3 | <0.0001 |
| | Cluster | 1 | 674.7 | <0.0001 |
| | Depth | 9 | 52.0 | <0.0001 |
| $(Al_o\text{-}Al_p)/Si_o$ | Intercept | 1 | 2814.2 | <0.0001 |
| | Cluster | 1 | 197.3 | <0.0001 |
| | Depth | 9 | 2.2 | 0.020 |
| $Al_o + 0.5Fe_o$ | Intercept | 1 | 754.3 | <0.0001 |
| | Cluster | 1 | 505.4 | <0.0001 |
| | Depth | 9 | 3.4 | 0.004 |
| Allophanes | Intercept | 1 | 636.7 | <0.0001 |
| | Cluster | 1 | 278.6 | <0.0001 |
| | Depth | 9 | 2.6 | 0.005 |

**Fig. 1.** Table 2

Table 3. Descriptive statistics for the predicted andic properties of all soil samples (All) and for those in spectral clusters *ALL* and *H*. "Depth" denotes the depth range from which the sample was taken. Entries are means ± standard deviation. Letters indicate significant differences among soil depth at p < 0.05.

[revised manuscript text omitted]

---

## Author Response (AR1)

**Topical Editor Decision: Reconsider after major revisions** (15 Jul 2019) by Karsten Kalbitz

Thanks a lot for your comments in our manuscript. We did not upload a marked-up manuscript version in the plateform as there too many changes. Specific answers to your comment are listed below. Editor's comment in black, answers in blue.

1/ According to WRB there are 3 diagnostic criteria for andic properties (an Alox + ½Feox value of ≥ 2%; and a bulk density of ≤ 0.9 kg dm-3; and a phosphate retention of ≥ 85%.). The authors should comment on that. Furthermore, they have to develop a convincing strategy for using proper terms. The replacement of andic properties by andic soil materials is not the best option because the latter is not really defined. The alternative provided in the response letter "short-range-order constituents" might be used consistently and throughout the manuscript. However, please replace "constituents" by "minerals".

As we did not have data for phosphate retention, we did not especially comment on the 3 diagnostic criteria proposed. We used the term *allophanic* soil material for the soil material which have $Al_o + 0.5 Fe_o \gg 2\%$ (L. 240-248) bulk densities < 0.8 (L.389) and the term halloysitic soil material for the soil material which have $Al_o + 0.5 Fe_o \leq 2\%$ (L. 240-248) and bulk densities > 0.8 (L.389).

**Modifications in the text** : L. 240-248 : *"Allophanic* soil materials, corresponding to the soils in the *allophanic* spectral cluster, rich in organo-Al complexes ($0.42 \pm 0.12$ g $Al_p$ 100 $g^{-1}$ soil), and SRO minerals: allophane ($15.8 \pm 4.4$ g allophane 100 $g^{-1}$ soil), $Al_o + 0.5 Fe_o$ ($5.3 \pm 1.2$ g $100g^{-1}$ soil) and amorphous Al, Si and Fe ($4.5 \pm 1.1$ g $Al_o$ 100 $g^{-1}$ soil, $1.6 \pm 0.4$ g $Si_o$ 100 $g^{-1}$ soil, $1.5 \pm 0.2$ g $Fe_o$ 100 $g^{-1}$ soil). These soils have $Al_p:Al_o$ ratios of about $1.0 \pm 0.4$, and Al:Si ratios of about $2.6 \pm 0.2$. In contrast, *halloysitic* soil materials, corresponding to the soils in the *halloysitic* spectral cluster are poor in organo-Al complexes ($0.18 \pm 0.11$ g $Al_p$ 100 $g^{-1}$ soil), and SRO minerals: allophane ($5.3 \pm 2.9$ g allophane 100 $g^{-1}$ soil), $Al_o + 0.5 Fe_o$ ($1.9 \pm 0.9$ g $100g^{-1}$ soil), and amorphous Al, Si and Fe ($1.5 \pm 0.8$ g $Al_o$ 100 $g^{-1}$ soil, $0.7 \pm 0.3$ g $Si_o$ 100 $g^{-1}$ soil, $0.9 \pm 0.3$ g $Fe_o$ $kg^{-1}$ soil). Their $Al_p:Al_o$ ratios are highly variable (about $1.7 \pm 1.5$), and their Al:Si ratios are about $1.7 \pm 0.6$. In a given cluster, the $Al_p$ and the $Fe_o$ content, the $Al_p:Al_o$ ratio decrease with depth, but not the allophane content, Al:Si ratio, nor the quantity $Al_o + 0.5 Fe_o$. "

We replaced "constituents" by "minerals" as proposed and used more consistently as previously "SRO minerals".

2/ The whole manuscript is by far too long. Particularly the discussion is sometimes very broad and not all the time focused on the main hypotheses of the paper. One reason for that might be the combined "Results and Discussion" section. The authors should think about the advantages to change the whole structure of the manuscript (separated results and discussion sections). I give you some examples in the manuscript where the need for reduction is very obvious to me: lines 277-285, 328-339, 353-378, 401-415, 436-467, second part of the section 468-490

We did not separate results and discussion in two parts, but we shortened the discussion to focus on the main hypotheses of the paper. The number of lines were not dramatically smaller in that new version because of more extended abstract and material and methods sections as requested by the two previous reviewers.

4/ The acronyms for the two mineral clusters are not very straightforward. Using ALL for allophanic but also for all samples is misleading and a potential source of confusion. Do you really need an acronym? I would suggest to use the full names or more appropriate acronyms. There is no need to use these acronyms in tables – there is sufficient space for the full name.

We deleted the acronyms and replace by the full name *allophanic*, *halloysitic* and *allophanic-halloysitic* (the latter for the soil profile type ALL-H) in the masnuscript, figure, table and supplementary materials.

5/ In the Introduction (line 61), andic properties were used to explain high SOC stocks. I would skip "andic properties" here because the processes responsible for high SOC stocks are given.

OK we have skipped "andic properties".

6/Material and Methods: Estimation of andic properties: I would suggest to start with the most important criteria according to soil classification (e.g. Alo + 0.5 Feo) Line 433-434: oxalate extractable Fe is not equal to ferrihydrite

We have modified the section presenting "andic properties" as requested (L.178-188). As oxalate extractable Fe is not equal to ferrihydrite, we modified the discussion on $Fe_o$:

**Modifications in the text:** L.357-340 "In our study, we did not explicitly analyse the form of Fe extractable by oxalate, whether ferrihydrite dominated Fe forms or not, but the preservation of SOC related to $Fe_o$, even in relatively small amounts (0.9-1.3 g $Fe_o$ $100g^{-1}$ soil), seemed to be the major factor explaining SOC-content variations at the surface and at depth (Fig. 5 and Fig. 6)."

7/I suggest to reduce the number of tables and figures of the main manuscript. Please transfer some information into the electronic appendix (e.g. table 2?, table 3?, ??). There is no need for 11 figures in the main manuscript.

We reduced the number of tables from 8 to 6 and the number of figures from 11 to 8.

---

## Author Response (AR2)

**Topical Editor Decision: Publish subject to minor revisions (review by editor)** (05 Sep 2019) by Karsten Kalbitz

Comments to the Author:
Thank you very much for the revision you did. You really improved the manuscript a lot and there are just some minor issues which need to be addressed. I added my comments directly to the pdf files of the manuscript and the supplement. I will send you the supplement in an extra email because the websystem of the journal does not allow to attach two files.

Answer :

Thank you for your interest in our work and your minor comments to improve our the manuscript. Ichanged as you suggested the minor corrections in the text and in the supplementary materials. They are highlighted in red.

I still think that Fig 2A is important to understand what we have done. I have changed the figure caption as you suggested. Please tell me if it is clearer.

I also added two sections before the acknowledgements as requested "author contribution" and "competing interests". The references were carefully checked.

Sincerely,

Tiphaine Chevallier, on behalf of the authors